


# A two-folds deep learning strategy to correct and downscale winds over mountains.

Louis Le Toumelin[1], Isabelle Gouttevin[1], Clovis Galiez[2], and Nora Helbig[3, 4]

[1]Univ. Grenoble Alpes, Université de Toulouse, Météo-France, CNRS, CNRM, Centre d'Études de la Neige, Grenoble, France
[2]Univ. Grenoble Alpes, CNRS, Grenoble INP*, LJK, 38000 Grenoble, France *Institute of Engineering Univ. Grenoble Alpes
[3]WSL Institute for Snow and Avalanche Research SLF, Davos, Switzerland
[4]Eastern Switzerland University of Applied Sciences, Rapperswil, Switzerland

**Correspondence:** Louis Le Toumelin (louis.letoumelin@gmail.com)

**Abstract.** Assessing wind fields at a local scale in mountainous terrain has long been a scientific challenge partly because of the complex interaction between large-scale flows and local topography. Traditionally, the operational applications that require high resolution wind forcings rely on downscaled outputs of numerical weather predictions systems. Downscaling models either proceed from a function that links large scale wind fields to local observations (hence including a corrective step), or use

operations that account for local scale processes, through statistics or dynamical simulations, and without prior knowledge of large scale modeling errors. This work presents a strategy to first correct and then downscale the wind fields of the numerical weather prediction model AROME operating at 1300 m grid spacing, by using a modular architecture composed of two artificial neural networks and the DEVINE downscaling model. We show that our method is able to first correct the wind direction and speed from the large scale model (1300m), and then accurately downscale it to a local scale (30m) by using the DEVINE

downscaling model. The innovative aspect of our method lies in its optimization scheme that accounts for the downscaling step in the computations of the corrections of the coarse scale wind fields. This modular architecture yields competitive results without suppressing the versatility of the downscaling model DEVINE, which remains unbounded to any wind observations.

## 1  Introduction

Understanding the declination of synoptic winds at a local scale in complex terrain is crucial for a wide range of applications,

including assessing the dispersion of pollutants, predicting wildfire spread, and evaluating wind energy potential (Giovannini et al., 2020; Wagenbrenner et al., 2016; Dujardin and Lehning, 2022). Local winds also have a significant impact on the evolution of the snowpack. The high variability of wind fields in complex terrain generates local gradients in the surface energy balance, which in turn influence the interaction between the snowpack and the atmosphere. These interactions can lead to significant spatial variability in the seasonal snowpack at the slope scale (Mott et al., 2018). In addition, wind can cause

snow redistribution on snow-covered areas through erosion and deposition processes, which is a major concern for avalanche hazard prediction (Lehning and Fierz, 2008).



Wind fields variability at a local scale in mountains is largely driven by two factors: terrain forced-flow, which refers to the direct impact of topography on large scale winds, and thermally driven flows, which result from local temperature gradients caused by terrain inhomogeneity and variable shading (Whiteman, 2000). Terrain forced flow and thermal winds interact with

each other, causing local variations both in speed and direction, making it challenging to understand and model mountain winds.

Many applications rely on the ability of Numerical Weather Prediction (NWP) systems to model synoptic scale wind fields above mountains (Quéno et al., 2016; Vionnet et al., 2016). NWP are generally characterized by their horizontal grid spacing on the order of one or several kilometers. Despite constant increase in horizontal resolution in recent years, a large number of

use cases still require downscaling techniques to reach their resolution of interest (Vionnet et al., 2021; Marsh et al., 2020).

Several methods have emerged to adapt the wind fields provided by NWP systems (in this work referred to as "large scale") to a local scale. Statistical downscaling is a family of methods that adapt large scale information, such as NWP outputs, to local scale specificities using statistical operations. Another approach, dynamical downscaling, relies on models to directly simulate atmospheric and surface processes at higher resolution. A large variety of statistical downscaling methods can be found in the

literature: e.g. Dupuy et al. (2021) and Goutham et al. (2021) develop statistical downscaling methods specifically tailored to operate at their calibration sites. In a different way, Zamo et al. (2016) and Höhlein et al. (2020) adapt large scale NWP wind fields to specific target grids at a higher resolution. On the contrary, more general methods such as Winstral et al. (2017) can theoretically be applied to any area with the inclusion of appropriate terrain descriptors as inputs. Not only these methods increase the resolution of the simulated variables, but they also include corrective terms that can compensate for systematic

errors in NWP modeling: this is a direct consequence of the use of an optimization/training step that links modeled data to wind observations. Such methods can also be referred to as Model Output Statistics (MOS) or bias correction methods, and frequently present favorable statistics when evaluated using observed wind data. Since statistical relationships are derived by linking outputs from a specific NWP system to observed values, two challenges emerge. First, their use is restricted to a unique NWP system or NWP system version. Second, the model capability to extrapolate wind values on areas where no calibration

has been performed can be challenging and must be rigorously assessed.

Conversely, other downscaling methods restrict their use to the modeling or parameterization of local scale processes only, without any optimization based on observations. These methods may improve evaluation metrics through the added value of the representation of missing processes, however, they don't compensate for systematic errors in large scale modeling and hardly compete in terms of evaluation metrics with methods including a corrective step. However, their use is not restricted

to any specific NWP nor to any specific geographic area. A large array of the aforementioned models can be found in the literature, ranging from simple statistical relationships (Liston and Elder, 2006; Helbig et al., 2017) to dynamical downscaling methods including atmospheric models of various complexity (Wagenbrenner et al., 2016; Raderschall et al., 2008; Vionnet et al., 2017). DEVINE (Le Toumelin et al., 2022) is a brand-new example of statistical downscaling models that represent wind fields at a local scale without incorporating any fit to observed data. Indeed, DEVINE simulates the adaptation of large scale

wind fields to high resolution terrain (30m) by using a fully convolutional neural network. More specifically this model was trained at replicating the behavior of the atmospheric model ARPS over complex Gaussian topographies (Helbig et al., 2017).





Consequently, systematic errors originating from the NWP large scale inputs can eventually be transferred and amplified through DEVINE. These errors can have a variety of origins, like missing or incorrect parameterizations, too coarse model topography, errors due to the assimilation procedure, imprecise initial conditions... Furthermore, the use of a downscaling model also increases uncertainty in error attribution, making it difficult to determine whether the downscaled data reflects correct or incorrect simulations of local-scale processes or if error compensation scrambles model evaluation. However, even though error attribution is complex, identifying typical weather and topographic situations where inputs or downscaled data are incorrect is more accessible, notably thanks to deep learning.

As an illustration, Le Toumelin et al. (2022) observed that AROME wind fields are frequently underestimated at elevated and exposed areas. After using DEVINE, they noted lower errors: thanks to the ability of the downscaling model to simulate terrain forced flow at a local scale and notably strong wind accelerations over summits and crests, the initial NWP underestimation is reduced. Since some wind speed underestimation remains, it is not clear whether the downscaling model does not accelerate sufficiently the input wind, or if the initial NWP wind speeds are too low. Whatever their origin, deep learning technics jointly with in situ measurements and ancillary atmospheric and toporaphic data may enable an a posteriori compensation of such systematic biases.

In this context, we design and present a strategy, based on deep learning, that corrects NWP input wind fields upstream of DEVINE downscaling method. Indeed, the correction is made before the downscaling step, but the effect of downscaling is accounted for in the calculation of the correction. In turn, most errors affecting the coarse-scale wind fields are corrected without affecting the spatial extrapolation capabilities of the downscaling model and diminishing the associated performances. By scrutinizing a set of variables including many variables that can influence air motion (e.g. temperature, humidity, boundary layer height...) and advanced topographic metrics, the artificial neural networks developed for this correction optimize NWP wind speed and direction before calling the downscaling model. With this modular architecture, we provide an end-to-end chain including downscaling and model output statistics, that permit to boost the evaluation performances of the DEVINE downscaling model.

## 2  Data

In this study, we used forecasts from the AROME NWP system as inputs to our new downscaling strategy. We rely on forecasts from AROME both for large scale wind fields and other atmospheric variables used in the corrective step. Our models also make use of high resolution topographical information (30m). Quality-controlled wind observations acquired over a large network of automatic weather stations (AWS) are used for model training (train set) and evaluation (test set). We finally compared the performance of our models to the operational analysis of the AROME system.

### 2.1  AROME

The AROME NWP system embeds a limited-area model, notably run by Meteo-France for short term weather forecasting operations. It simulates the state of the atmosphere and the surface over a European domain including the french Alps, Pyrenees





and Corsica. The model solves the non-hydrostatic fully compressible Euler Equations by using a semi-Lagrangian and semi-
implicit numerical solver and by including a spectral representation of several prognostic variables (Seity et al., 2011; Bénard
et al., 2010). The physics is inherited from the Meso-NH model (Lafore et al., 1998; Lac et al., 2018) and the dynamical core
from ALADIN-NH (Bubnová et al., 1995). The model is driven at its borders by the ARPEGE model. It simulates energy
and mass exchanges between the atmosphere and the surface thanks to the SURFEX platform (Masson et al., 2013). Notably,
AROME uses the SURFEX/ISBA model over land (Noilhan and Mahfouf, 1996; Masson et al., 2013) and the simplified
snowpack scheme from Douville et al. (1995) over snow covered areas. Since 2018, AROME operates with a 1.3km horizontal
grid spacing over France, which is of high interest for applications that require high resolution information about the state of
the boundary layer such as weather forecasting over complex terrain (Quéno et al., 2016; Vionnet et al., 2016). The AROME
system also includes a 3Dvar assimilation scheme, which takes into account radial winds observed by radars in addition to the
assimilation of 10-m wind speeds. We note that wind observations in complex terrain are frequently neglected for assimilation
due to their lack of spatial representativity (Gouttevin et al., 2023). Eventually, their distance to AROME initial guess can also
lead to their exclusion of the assimilation cycle.

AROME analyses are produced every UTC hours whereas the model is also run in forecast mode every 3h. For this study, we
built two different products from the aforementioned cycles. Firstly, we built a continuous time series by extracting +6 to +29h
AROME forecast lead times, initialized with the analysis of 00:00 UTC, as in Quéno et al. (2016). This way, we were able to
construct a continuous time series of 11 variables from AROME forecasts between the $1^{st}$ of September 2017 to the $1^{st}$ of
October 2020 at an hourly time step (AROME$_{forecast}$). The variables are detailed in Table 2 and their respective use described
in Sect. 3.3. For the same period, we extracted the same variables, from the analysis cycles (AROME$_{analysis}$) also at an hourly
time step. Finally, we dispose of two datasets from AROME: AROME$_{forecast}$ are representative of forecasted atmospheric
and surface conditions and AROME$_{analysis}$ are representative of an a posteriori product, giving the most plausible state of
the atmosphere at the considered date. In the following study AROME$_{forecast}$ are used as inputs of the post-processing and
downscaling schemes, as would be within an operational high-resolution forecast system, whereas AROME$_{analysis}$ serves as
a reference "best" product to compare with.

## 2.2  Observations

We gathered hourly wind field observations from automatic weather stations originating from different observation networks in
Switzerland and France in order to train and evaluate our models. In detail, we used a total of 273 observation stations. Among
them, 214 stations are located in Switzerland, and correspond to data provided by MeteoSwiss, the Swiss Federal Office of
Meteorology and Climatology. Then, 59 stations are located in France among which 54 are from Météo-France observational
networks and 5 from the GLACIOCLIM network ("Les GLACIers un observatoire du CLIMat" - "Glacier: an observatory of the
climate"). We note the use of three AWS from the Col du Lac Blanc instrumental site, a high-altitude observatory specifically
dedicated to the study of mountain meteorology and drifting-snow (Vionnet et al., 2017; Guyomarc'h et al., 2019). We used
218 stations to train neural networks and 55 distinct stations for model evaluation (see Sect. 3.4 for the sampling strategy). The

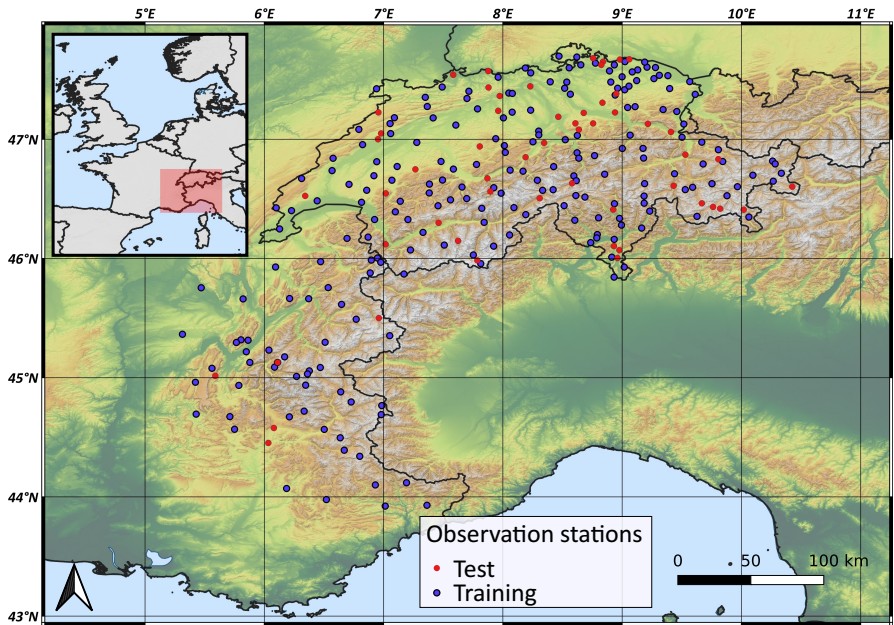

**Figure 1.** in situ observation stations used to train the model (218 blue dots) and evaluate the model (55 red dots). All stations are localised in Switzerland and France. Train/test partitioning was performed using a stratified sampling method described in Sect. 3.4. Note that an additional data set constituted from observation collected in Corsica and in the french Pyrenees has been used in Sect. 4.5 in order to evaluate the applicability of our model to other mountain ranges.

observational sites are located in various types of environments, all representative of alpine terrain. This includes snow covered areas, slopes, exposed terrain but also lower elevation valleys and some stations localized around urbanized terrain.

Since most of the wind observations used in this study were obtained in complex terrain and frequently under challenging
meteorological conditions, we applied a quality check procedure to our observational dataset, inspired from Lucio-Eceiza et al. (2018a, b). As extensively detailed in Le Toumelin et al. (2022), this procedure first asserts a correct data compilation and storage including chronological sorting, a search for eventual repeated dates and the distinction between true North (360°) and undefined direction when wind speed is null (0°). Moreover, it ensures the validity of observed speeds and directions by removing unrealistic observations (e.g. speeds > 100 $m\,s^{-1}$ or negative directions). The quality check also includes the use of
log profiles to unify the observational height of wind fields (here set at 10m) for measuring devices located below 10m above the surface. This procedure uses snow height information when available to adjust wind profile correction. Moreover, some additional tests are designed to detect suspicious speeds or direction sequences. As an example, icing of wind sensors is a typical case of sensor dysfunction in complex terrain and is reflected by the acquisition of null or constant speed/direction for several consecutive hours. The quality check procedure also takes care of other typical suspicious sequences of data such as extremely
high speed variations (extreme spikes or lows in time series) and constant sequences of positive speeds for consecutive hours or days. Finally, longer term rolling means are scrutinized to detect the suspicious rise or decline of observed mean speeds





which can shed light on the occurrence of systematic errors of diverse nature (e.g. mast tilting, new vegetation or urbanization in the vicinity of the sensor).

### 2.3 Terrain parameters

Since the local topography has a large impact on wind fields, several topographic parameters are used as input variables to the corrective strategy, so as to capture the dominant local features of the topography. Among the selected parameters, the $TPI_{500m}$ (Weiss, 2001) consists of computing the difference between a digital elevation model (DEM) pixel elevation and the mean elevation of neighboring pixels within a fixed radius (here taken at 500m). Consequently, the $TPI_{500m}$ gives an integrated vision of the relative elevation of the considered pixel: positive $TPI_{500m}$ indicates that the pixel of interest is higher than the

neighboring pixels, negative $TPI_{500m}$ indicate the opposite. The curvature, computed following Liston and Elder (2006), quantifies how much a terrain differentiates from a plane. The Laplacian, computed as in Le Toumelin et al. (2022), also gives an estimation of the local elevation variation and enables to detect small scale peaks or bowls within topographic maps. The slope, obtained as the root mean squared slope using first order finite differences as in Helbig et al. (2017), quantifies the local slope of topography. The aspect indicates the orientation of a pixel relative to the North direction. Finally, the parameter $\alpha$,

adapted from Dujardin and Lehning (2022) is computed following equation 1.

$$\alpha = \arctan(\tan(\text{slope}) * \cos(\text{wind direction} - \text{aspect})) \tag{1}$$

$\alpha$ is a proxy firstly indicating how wind direction should be modified in order to align perpendicularly to the aspect. Furthermore $\alpha$ also increases with the slope so that it is higher over steep slopes than over flat terrain. Similarly to wind direction, $\alpha$ is expressed in degree. Since $\alpha$ and aspect are computed using values from direct neighboring pixels, they tend to be sensible

to small scale variations in topographic features. To reduce this variability, we averaged all aspects and $\alpha$ values using a 3x3 moving window, i.e. averaging all $\alpha$ values given the 8 $\alpha$ values from the neighboring DEM pixels (30m spaced).

### 3 Method

### 3.1 Artificial neural network

Artificial neural networks (ANNs) are a specific type of machine learning model. They are constituted by interconnected units

called neurons, which hold floating point values, all organized in different layers. In a layer, neurons transmit the information received from previous layer's neurons to the next layer's neurons. Communication consists of first an affine modification of each neuron value using weights (slope parameters) and biases (intercepts). Then, all neuron modified values are summed and pass through a nonlinear activation function which produces the next layer's neuron input values. Finally, the first layer holds the raw inputs, while the last layer holds the predicted values. All weights and biases are typically initialized using random

values and then modified using optimization algorithms based on gradient descent methods. Such methods are based on the computation of the gradient of a loss function between the neural network output and the expected output (ground truth) with





respect to the network weights and biases. Weights and biases are then optimized in the opposite direction of the gradient in order to minimize the loss. By replicating this strategy a large number of time over a large number of samples, artificial neural networks can learn complex patterns that link the training inputs to the training target outputs. Finally, we note the existence

of different hyperparameters, which consist of parameters that are not weights and biases (e.g. the number of neurons and the number of layers). These parameters are not learnt during the training process but rather fixed independently.

## 3.2  DEVINE

DEVINE is a downscaling model based on a Unet convolutional neural network (Ronneberger et al., 2015) designed to adapt wind fields to high resolution topography (30m) in complex terrain (Le Toumelin et al., 2022). This model takes as inputs high

resolution topography (30m) and large scale wind fields above the topography and provides 3D wind fields with a 30m grid spacing as output. DEVINE uses convolutions to detect advanced spatial features on topographic maps and to assemble them into more complex patterns within a latent space. This latent representation is then used to reconstruct high resolution wind fields using spatial interpolation and convolutions to finally obtain wind simulations at the same resolution as the input topography. The model was trained using 7279 ARPS simulations performed by Helbig et al. (2017). These simulations were run over

a large range of synthetic Gaussian topographies of diverse complexity, using similar constant initial atmospheric conditions across all simulations. The trained model showed good behavior at reproducing ARPS simulation on an evaluation dataset. As a case study, Le Toumelin et al. (2022) applied DEVINE to downscale AROME forecasted wind fields in the french Alps and used observations from 61 in situ stations for model evaluation. Qualitatively, the model simulates coherent spatial structures, characteristic of terrain forced flows. Notably, the model is able to detect ridges and summits and to simulate acceleration

as observed within ARPS simulations that served as labels. Similarly, DEVINE shows good behavior at detecting windward and leeward areas and is able to modify speed accordingly. Some directional shifts were observed around topographic barriers (channeling), but remain modest. Some other features of mountain winds occurring at the slope scale such as recirculation areas or upslope/downslope thermal flows are not accounted for. Quantitatively, in addition to its spatial extrapolation abilities, DEVINE improves AROME evaluation metrics notably at the most elevated and exposed stations. A significant improvement

in modeling the highest wind speeds has also been observed, which is of high interest for applications demanding good precision above a certain speed threshold such as drifting-snow modeling. In its previous technical implementation, DEVINE used a combination of pre-processing, downscaling and post-processing operations which were partly written using the Tensorflow framework (Abadi et al., 2015), partly using other python libraries. In it's current version, DEVINE is fully written using Tensorflow operations, permitting gradient error backpropagation through all the architecture of the model (and not only through

its Unet part), which is necessary to construct the new model presented in Sect. 3.3.




**Table 1.** Input variables used in ANN$_{speed}$ and ANN$_{direction}$

| Name | Unit | ANN$_{speed}$ | ANN$_{direction}$ | Details |
|---|---|---|---|---|
| $Wind\,speed_{5m}$ | $m\,s^{-1}$ | x | | |
| $Wind\,speed_{10m}$ | $m\,s^{-1}$ | x | x | |
| $Wind\,speed_{50m}$ | $m\,s^{-1}$ | x | | |
| $Wind\,speed_{126m}$ | $m\,s^{-1}$ | x | | |
| $Wind\,speed_{515m}$ | $m\,s^{-1}$ | x | | |
| $Wind\,direction$ | ° | x | x | 10m wind direction |
| $Boundary\,Layer\,Height$ | $m$ | x | | |
| $LW_{net}$ | $W\,m^{-2}$ | x | | Longwave radiation budget |
| $SW_{net}$ | $W\,m^{-2}$ | x | | Shortwave radiation budget |
| $T_{2m}$ | $°C$ | x | | 2-m temperature |
| $Cloud\,cover$ | | x | | Varies between 0 (no cloud) and 1 |
| $Elevation_{model}$ | $m$ | x | | AROME surface elevation |
| $Elevation$ | $m$ | x | | Elevation from a 30m DEM |
| $TPI_{500m}$ | $m$ | x | | |
| $Laplacian$ | $m^{-1}$ | x | | |
| $Slope$ | | x | | |
| $Curvature$ | | x | | |
| $\alpha$ | ° | | x | see Sect. 2.3 |
| $aspect$ | ° | | x | |

## 3.3 Neural network + DEVINE

### 3.3.1 Architecture

The model presented in this study corresponds to an extension of the DEVINE model. It consists of the addition of two ANNs that process large scale NWP data and local scale topographic data prior to the use of the DEVINE downscaling model. More precisely, a first neural network is designed to compute an additive correction for the NWP wind direction (ANN$_{direction}$) aiming at compensating for large scale modeling errors, and a second network performs similar corrections for the NWP wind speed (ANN$_{speed}$). The modified large scale wind speed and direction are then used to feed the DEVINE downscaling model, that also uses a high resolution topographic map (30m) of the area considered. In details, ANN$_{direction}$ uses 4 input variables (2 topographic parameters and 2 variables from the NWP system) and ANN$_{speed}$ uses 17 variables (5 topographic variables and 12 NWP variables), all listed in Table 2. The outputs of the overall model (referred to as *Neural networks + DEVINE*) are the same as DEVINE outputs, i.e. high resolution maps of the three components of the wind vector.





**Table 2.** Architecture, hyperparameters ans loss functions used in Neural Network and Neural Network+DEVINE

| Name | $\text{ANN}_{speed}$ | $\text{ANN}_{direction}$ | Details |
|---|---|---|---|
| Activation function | Selu | Gelu | Excluding output neuron |
| Activation function (output) | Linear | Linear | Output neuron only |
| Batch size | 128 | 128 | |
| Dropout rate | 0.25 | 0.35 | |
| Epochs | 10 | 5 | |
| Initializer | GlorotUniform | GlorotUniform | Glorot and Bengio (2010) |
| Learning rate | 0.001 | 0.001 | |
| Loss function | $\mathcal{L}_{speed}$ | $\mathcal{L}_{direction}$ | Sect. 3.3.3 |
| Number of layers | 2 | 2 | Excluding input and output layers |
| Optimizer | Adam | Adam | Kingma and Ba (2014) |
| Units per layer | [50, 10] | [50, 10] | |

As the $\text{ANN}_{direction}$ and $\text{ANN}_{speed}$ need to output wind speed and direction, they need to take into account the typical range of wind speed (positive values, generally below $100\ m\,s^{-1}$) and direction values (0 to 360°). To facilitate such a task, we used skip connections: considering $\text{ANN}_{direction}$ (respectively $\text{ANN}_{speed}$), the initial NWP direction (respectively speed) is added to the value of the ANN's output neuron so that the network concentrates on the computation of a direction difference (respectively speed difference) instead of computing directly direction (respectively speed). Furthermore, care has to be taken to activation functions used before the skip connection: the direction difference (respectively speed difference) should not be constrained to positive or negative values only (as in relu functions for instance) since modifications can be either positive or negative depending on weather and topographic situations. Hence, we selected a linear activation function for the last layer of both input networks, before calling the skip connection layer. Furthermore, after adding modifications suggested by the network to the initial wind direction (respectively speed), i.e. after the skip connection layer, we had to ensure that no negative values were produced. For that, we used a relu activation function that caps negative values to zero. Hyperparameters and architecture details are summarized in Table 2.

### 3.3.2 Training

In order to adapt the weights and biases of the ANNs, we adopted a sequential approach. First, we optimized $\text{ANN}_{direction}$ for wind direction, and then optimized $\text{ANN}_{speed}$ for wind speed. This order is motivated by the fact that an erroneous direction can translate into erroneous high-resolution wind speeds with DEVINE as a result of wrong topography adjustments, whereas the opposite will have less impact. We selected 218 training observation stations in the french Alps measuring wind speed and direction, and used topographic maps centered around each station. Additionally, we used data from the nearest grid cell of AROME at each of these stations to take into account large scale atmospheric conditions. Since our network outputs wind field





maps and we only dispose of localised observation, we selected the pixel at the center of each model's prediction (which hence coincides with the location of an observation station) and optimized the output of our network at these locations to match given in situ observations. The optimization process involved backpropagating the gradient of a loss function, which was computed using the wind direction or speed value simulated by DEVINE and the observed ground truth. The loss functions used in

this study are describe in Sect. 3.3.3 and correspond to a cosine distance for optimizations of the direction, and a modified mean squared error for optimization of wind speed. During the optimization of $ANN_{direction}$, both DEVINE and $ANN_{speed}$ weights and biases are kept frozen. Similarly DEVINE and $ANN_{direction}$ weights and biases are not updated during $ANN_{speed}$ optimization. We note that DEVINE parameters were directly taken from the original model Le Toumelin et al. (2022) and have not been modified in this study. This choice was made because our goal was to develop an optimization system to be

used with DEVINE, rather than creating a new downscaling model (see Sect. 5). Once trained, Neural network + DEVINE can model wind fields at high resolution, even over areas not included in the training process. Additionally, intermediate values (i.e. ANN outputs, referred to as *Neural Network*) are saved for model interpretability purpose (red dots in Fig. 2).

### 3.3.3  Loss functions

Two loss functions were selected for training $ANN_{direction}$ and $ANN_{speed}$. For $ANN_{direction}$ we selected the cosine dis-
tance ($\mathcal{L}_{direction}$, Eq. 2) to account for angular differences between direction predictions ($direction_{model}$) and observations ($direction_{obs}$). We also took care to express all directions in degrees or radians when required.

$$\mathcal{L}_{direction} = 1 - \cos(direction_{obs} - direction_{model}) \tag{2}$$

For $ANN_{speed}$ we designed a custom loss function that targets the main errors typically found in AROME forecasted wind fields. Previous studies (e.g. Dujardin and Lehning (2022); Bolibar et al. (2020)) demonstrated that the use of a classic loss
function (e.g. mean squared error) tends to produce squeezed distribution around the mean value of the output and poor evaluation metrics. Our loss function, denoted as $\mathcal{L}_{speed}$ (Eq. 3), is designed to penalize three specific characteristics of AROME's wind field errors as follows: $\mathcal{L}_{speed}$ (i) compares simulated values to actual in situ observations using the mean squared error ($mse$), (ii) uses the factor $\tau$ to foster the correction of speed underestimations over overestimations ($\tau$ is arbitrarily fixed to 0.6 for cases of underestimations and 0.4 for overestimations) and (iii) places a higher penalty on errors made at high wind speeds
by scaling $\mathcal{L}_{speed}$ with observed speeds ($speed_{obs}$).

$$\mathcal{L}_{speed} = speed_{obs} * \tau * mse(speed_{obs}, speed_{model})$$
$$\text{where} \quad \begin{cases} \tau = 0.6 & \text{if } speed_{obs} \le speed_{model} \\ \tau = 0.4 & \text{if } speed_{obs} > speed_{model} \end{cases} \tag{3}$$


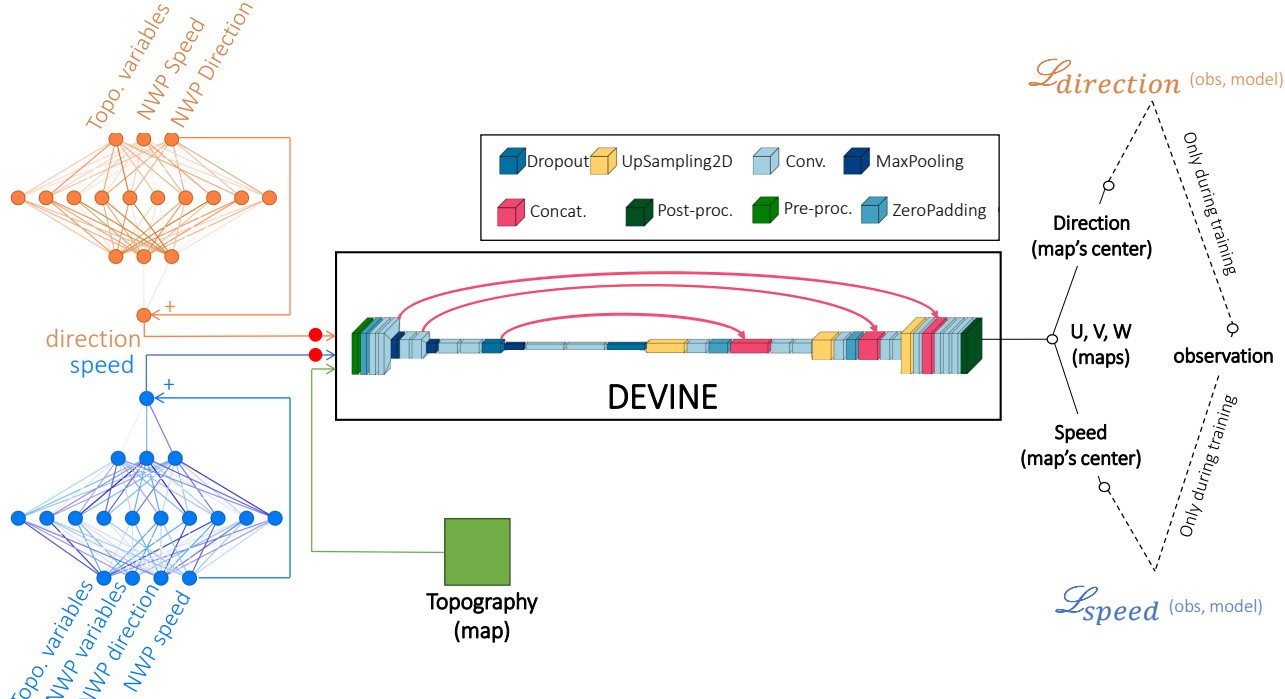

**Figure 2.** Scheme of the new model architecture. The architecture is composed of two artificial neural networks (ANN) in addition to DEVINE downscaling model. The first ANN predicts wind direction (orange, $\text{ANN}_{direction}$), the second one predicts wind speed (blue, $\text{ANN}_{speed}$). Modified wind speed, direction and a high resolution topographic map are then sent to DEVINE (Le Toumelin et al., 2022), which in turn outputs maps of the three components of wind fields (U, V, W) at high resolution (30m). During the training step, wind direction and speed values are computed at the maps' center (taken to coincide with an observation station) and sequentially compared to in situ observation using appropriate loss functions (see Sect. 3.3.3). $\text{ANN}_{direction}$ and $\text{ANN}_{speed}$ optimizations are guided by the gradient of these losses. We note that $\text{ANN}_{direction}$ and $\text{ANN}_{speed}$ have both an independent model architecture, including the nature of inputs variables. "Topo. variables" refer to input variables of topographic nature, "NWP variables" to inputs corresponding to forecasted meteorological and surface variables, "NWP direction" to forecasted wind direction and "NWP speed" to forecasted wind speed. "Dropout", "UpSampling2D", "Conv." (which stands for convolution), "MaxPooling", "Concat." (which stands for concatenation) and "ZeroPadding" correspond to mathematical operations performed within DEVINE (Le Toumelin et al., 2022). "Pre-proc." (which stands for pre-processing) and "Post-proc" (which stands for post-processing) correspond to operations that permit to use DEVINE CNN given any meteorological condition and notably include the rotation of the input topography to take into account initial wind direction in predictions. Finally, red dots following $\text{ANN}_{direction}$ and $\text{ANN}_{speed}$ consist of the intermediate results of the network (i.e. the ANN outputs) and are referred to as Neural Network.





### 3.4 Data partitioning

Deep-learning applications commonly involve the use of a training set for model optimization and a test set for model evaluation. Many studies (Goutham et al., 2021, e.g) implement random train/test split, i.e. randomly extracting test samples from

the training set to form a test set. As underlined by Dujardin and Lehning (2022), this method can lead to an overestimation of the model performance. Evaluating a model after random sampling in a temporal context is equivalent to assessing the ability of the model to reconstruct an incomplete time series given the information of all other known time steps. Furthermore, using random split or a simple temporal split means that the ability of the model to predict at unknown areas is not documented. This can be detrimental for a large number of applications that require downscaled data over areas different from the calibration

area. In this study, we decided to evaluate our model both over observational sites not used during training and for a year that was not included during training. This method corresponds to a spatio-temporal extrapolation assessment and provides a strict evaluation procedure, closer to real use cases where a model is run over diverse areas, largely not present in the training set. Consequently, we divided our dataset into a training set and a test set using a temporal and a spatial split.

**Space partitioning**

The spatial split involved a stratified selection process that resulted in the selection of 55 AWS sites from the 273 sites available in the Alps. We first identified six topographic and geographic descriptors for the AWS locations, calculated as described in Sect. 2.3 : elevation, the TPI, the slope, the local Laplacian, and the x and y geographical coordinates of the stations (expressed using the Lambert93 projection). For each parameter, we split the 273 AWS sites in 3 groups according to their position in the parameter's distribution: stations with a parameter below the 0.33 quantile, between the 0.33 and 0.66 quantile or above the

0.66 quantile. We then divided each of these three groups into three additional categories, according to the root mean square error (RMSE) of $AROME_{forecast}$ at each site. We applied a random sampling without replacement in the final three groups and ensured that no station was selected twice. Considering the six parameters categorized in three intermediate groups that are in turn categorized in 3 groups, we identify $6 * 3 * 3 = 54$ stations that are representative of diverse topographic parameters, geographic locations and AROME performances. We also included Col du Lac Blanc station (latitude = 45.12°, longitude =

6.11°; elevation = 2720 m), as it has been studied in Le Toumelin et al. (2022) and we wanted to study our new model at this site. After this spatial split, our training set is composed of the remaining 218 AWS sites and our test set of the 55 selected AWS sites. The stratified selection process favors the selection of a test set that is balanced among the six selected parameters and has a diverse range of AROME performance, limiting the risk of unbalanced properties of the observational sites among training and test sets.

**Time partitioning**

The temporal split simply consisted of excluding the last year of data from the training set, and excluding the first two years from the test set. Finally we dispose of two years of data at 218 sites for training and one other year at 55 other sites for evaluation.





### 3.5 Neural network interpretability

#### 3.5.1 Partial Dependence Plots

In statistical modeling, interpretability methods give insights on the causes that lead a model to make a specific decision. Among these methods, the Partial Dependence Plots (PDP) is an intuitive method giving insights on the isolated effect of a given variable on the model outputs. Their computation consist of iteratively fixing all instances of the studied input variable $variable_i$ at a precise value defined in a given range and observing the mean effect on the model outputs. By averaging over all model outputs, PDP permit to focus solely on the influence of $variable_i$ on the outputs. PDP suppose independence between input variables since fixing $variable_i$ to a given value comes with no modification of the other input variables. Using PDP with correlated features can lead to unrealistic situations where model predictions are performed for implausible data instances (e.g. studying the effect of temperatures > 20 °C over high altitude stations during winter nights). However contrary to Accumulated Local Effects (see Sect. 3.5.2), PDP do not suppose any ordering in the input variable, contrary to accumulated local effects (see Sect. 3.5.2). Following this property, we use PDP in this work to study the impact of $ANN_{direction}$ input features on wind direction simulations.

#### 3.5.2 Accumulated local effects

Accumulated Local Effects (ALE) also permit to study the influence of a given input variable on the model outputs. Unlike more common methods such as PDP or feature importance ranking (McGovern et al., 2019), ALE are robust to correlated structures in the input variables, which frequently occur in atmospheric sciences. In contrast to PDP, ALE compute differences of prediction for a small window around specific values of a given input variable $variable_i$, based on its conditional distribution. In details, this is done by firstly grouping $variable_i$ values in $n$ bins of identical number of instances (quantiles). For each bin, a difference of model predictions is obtained after fixing all instances of $variable_i$ to the uppermost value of the bin and subtracting predictions obtained after fixing the same instances to the lowermost values of the bins. This permits to overcome the correlation issue of PDP because prediction differences are only computed for data instances in the considered $variable_i$'s bin. This step can be interpreted as a computation of local gradients around a specific value of $variable_i$. The differences are then averaged to obtain the local effect of $variable_i$ for the considered bin. Local effects are then accumulated and centered across each bin to finally obtain ALE. This steps corresponds to an integration of the (averaged) local gradients and enables to represent the dependence of model outputs to $variable_i$ across its range. Similarly, two dimensional ALE plots can also be obtained to highlight the effects of the interaction of two features within the model, without considering first order effects. Two dimensional ALE plots are well suited to observe if two features interact within the model and help decomposing higher order causes that lead to model prediction. In this study, ALE are used to understand how input variables of $ANN_{speed}$ influence Neural Network + DEVINE simulations. More details about ALE can be found in Molnar (2022).





## 4  Results

### 4.1  AROME performance in the Alps

AROME$_{forecast}$ performances in simulating wind speed in complex terrain depends on the topography. Indeed, we compared AROME$_{forecast}$ outputs to observed wind speeds in Figure 3, for a three year period at an hourly time step and for all stations available in the Alps (training + test). We then analysed the influence of topography by grouping observation stations by their quartiles in both $TPI_{500m}$ and elevation distributions. We observe that AROME$_{forecast}$ are marked by a negative mean bias

at both elevated and high $TPI_{500m}$ stations. The joint effect of $TPI_{500m}$ and elevation is all the more marked since speed discrepancies increase with $TPI_{500m}$ for the highest elevation category. Oppositely, for lower elevation and $TPI_{500m}$ closer to 0 (i.e. $TPI_{500m}$ in the second and third quartiles), we note a positive speed bias, less intense than its negative counterpart. Numbers in Fig. 3 indicate the number of observation stations in each group and inform on the topographic characteristics of our observational dataset. Notably, we observe that elevated stations coincide with high absolute value of $TPI_{500m}$. High

positive values of $TPI_{500m}$ indicate that the observation station dominates its neighborhood, and is to some extent "exposed": a feature that occurs mainly in high elevation areas in our dataset. Similarly, most of the lowest elevation stations have $TPI_{500m}$ in the second and third quartiles. $TPI_{500m}$ close to zero characterize stations on average at the same elevation than their neighborhood in a radius of 500m, a definition that includes flat terrain.

Complementary, we observe that AROME$_{forecast}$ negative bias varies with observed wind speed. Figure 4 (a) compares

AROME$_{forecast}$ hourly simulations to hourly observations and shows the onset of a negative bias with increasing observed speed. This behavior is characterized by a departure from the 1-1 line for the highest observed wind speeds. This observation is consistent with Fig. 3 since generally, (i) wind speed increases with elevation, and (ii) high speeds are generally observed over summits, crests and ridges (Whiteman, 2000), which designate topographic features often characterized by high $TPI_{500m}$. Figure 4 confirms and generalises the results from Le Toumelin et al. (2022) who already evidenced this AROME$_{forecast}$

underestimation pattern in the french Alps: note that the test set used in Fig. 4 shares 5 observation stations with Le Toumelin et al. (2022) data set.

Finally, AROME$_{forecast}$ captures realistic wind direction patterns in the Alps. This is qualitatively shown in Fig. 5 (a) and (d) where AROME wind distribution closely resembles the observed wind distribution. We nevertheless observe discrepancies such as a shift in the most frequent wind direction. Indeed, the west-southwest wind direction is the most frequent direction

among our observations whereas AROME$_{forecast}$ predominantly simulates south-west wind fields. For all directions, we note that most wind direction errors are inferior to 60°, and inferior to 30° when forecasted among the dominant directions (west-southwest and south-west). The largest direction errors (i.e. errors superiors to 90°) affect all directions in comparable proportions. We finally observe that AROME$_{forecast}$ tends to overestimate the west-northwest, northwest and north-northwest direction while underestimating the north directions.

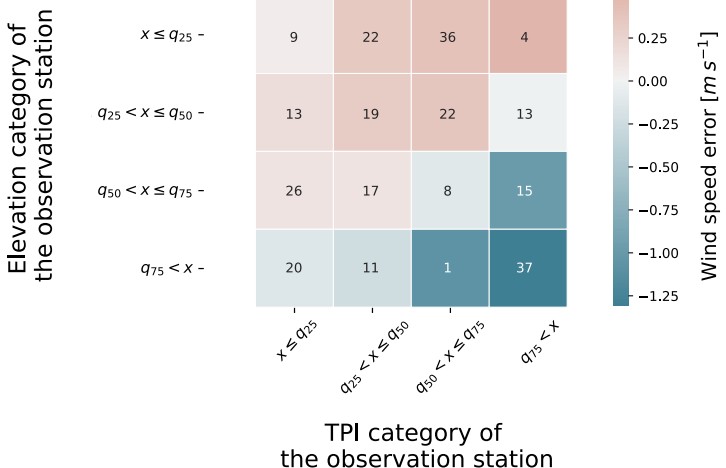

**Figure 3.** Mean wind speed error of AROME forecasts versus observed wind speed (color) at all stations available (training + test) in the Alps. The results are categorized by $TPI_{500m}$ and elevation quartiles ($q$) at the observation stations. The numbers indicate the number of observation stations within each category. $TPI_{500m}$ (respectively elevation) values of $q_{25}$, $q_{50}$ and $q_{75}$ are -20m, -2m and 11m (respectively 482m, 859m, 1605m).

## 4.2 Model evaluations

In this section, we evaluate the performances of different wind products, including AROME$_{forecast}$ and AROME$_{analysis}$ but also the results of our deep-learning corrections and/or downscaling models (DEVINE, Neural Network, and Neural Network+DEVINE). Consequently, we use the test dataset, which was not used to train the deep learning models. We remind that AROME$_{forecast}$ serves as input for DEVINE and Neural Network+DEVINE, while both deep learning models did not use directly any data from AROME$_{analysis}$ as input. Integrated evaluation metrics first highlight improved RMSE, MAE, mean bias and coefficient correlation with DEVINE over AROME$_{forecast}$ (Table 3). Such improvements are not able to bridge the gap between AROME$_{forecast}$ and AROME$_{analysis}$, the latest showing largely improved evaluation metrics. However, the use of Neural Network+DEVINE improves statistics (except mean bias), ultimately showing the best results among all wind products.

We also observe (as expected) an improved behavior of AROME$_{analysis}$ over AROME$_{forecast}$ notably through a partial correction of the departure from the 1-1 line for high observed speeds initially observed in AROME$_{forecast}$ (Fig. 4). More generally, AROME$_{analysis}$ data are centered around the 1-1 line suggesting a better agreement between simulations and observations. Similarly, we observe that DEVINE generates increased wind speed, notably for the highest observed speeds. Such a modification compensates for AROME$_{forecast}$ initial underestimation. However, contour lines which indicate data density still reveal some dispersion around the 1-1 line with DEVINE. Neural Network+DEVINE also shows a partial correction for the highest observed speeds and shows generally less dispersion around the 1-1 line. A close inspection of the lowest wind speeds however indicates some overestimation of null and speeds inferiors to $1\,m\,s^{-1}$.





We then scrutinized the model performances for wind speed with respect to elevation in Fig. 6. DEVINE performances are comparable to AROME$_{forecast}$ performances for low elevated stations, additionally to the fact that contrary to AROME$_{forecast}$, DEVINE provides a spatialized signal at a local scale. Improvements are however observed for higher stations and can be at-
tributed to the ability of DEVINE to simulate acceleration at exposed and elevated stations where AROME$_{forecast}$ denotes a negative bias compared to observations. These results reinforce previous study with DEVINE (Le Toumelin et al., 2022), who observed similar behaviors. AROME$_{analysis}$ presents better evaluation metrics compared to AROME$_{forecast}$ and DEVINE at all elevations categories. However, we still observe some errors at the most elevated stations. Neural Network+DEVINE finally improves DEVINE evaluation metrics at all elevations categories, matching AROME$_{analysis}$ metrics on the second and third
quartiles and outperforming it for the most elevated stations. In detail, the boxplot indicates slightly lower median errors for AROME$_{analysis}$ compared to Neural Network+DEVINE at all categories except for the highest stations, but also shows that the largest modeling errors are less frequent with Neural Network+DEVINE among the third and fourth quartile.

In terms of wind direction, AROME$_{analysis}$ largely diminishes the largest modeling errors observed in AROME$_{forecast}$. Wind distribution patterns highlight a reinforcement of the occurrence of wind in the south-west direction, which is still differ-
ent from the observed wind patterns (Fig. 5) However, we see improvement in the reduction of north north-west predictions and better characteristics concerning the North to East winds. On the other hand, as noted in Le Toumelin et al. (2022), DEVINE simulates directions close to AROME$_{forecast}$, without introducing any major change. Similarly to observations, Neural Network+DEVINE simulates most winds in the west south-west direction, and largely reduces the occurrence of the largest wind direction errors (Fig. 5). The improved performance is striking in the dominant westerly directions. Figure 6 sheds light on the
distribution of errors according to the elevation category of the observation stations and shows similar characteristics to speed errors. Similarly to AROME$_{analysis}$, Neural Network+DEVINE improves wind direction modeling over AROME$_{forecast}$ and DEVINE at all elevation categories, notably at the most elevated stations where Neural Network+DEVINE has the lowermost median value for direction error among all the products compared.

**Table 3.** Evaluation metrics obtained on the test dataset (Alps). MAE designates the mean absolute error, RMSE the root mean square error and $\rho$ the Pearson correlation coefficient. The mean absolute error for wind direction was computed by taking care of the cyclic nature of wind direction.

| Variable | Metric | AROME$_{forecast}$ | DEVINE | Neural Network | Neural Network+DEVINE | AROME$_{analysis}$ |
|---|---|---|---|---|---|---|
| Speed | MAE [$m\,s^{-1}$] | 1.34 | 1.29 | 1.21 | **1.16** | 1.18 |
| | RMSE [$m\,s^{-1}$] | 1.92 | 1.81 | 1.73 | **1.62** | 1.71 |
| | Mean bias [$m\,s^{-1}$] | -0.14 | **-0.02** | -0.17 | -0.05 | -0.15 |
| | $\rho$ [] | 0.60 | 0.66 | 0.68 | **0.72** | 0.69 |
| Direction | MAE [°] | 44 | 43 | **35** | **35** | 37 |



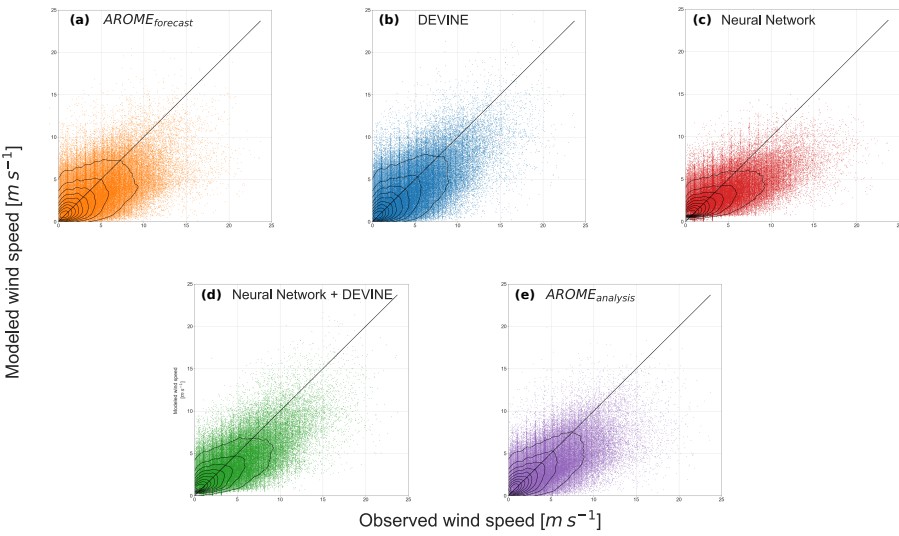

**Figure 4.** 1-1 plots of simulated versus observed wind speed. The models are (a) AROME$_{forecast}$, (b) DEVINE, (c) Neural Network, (d) Neural Network + DEVINE and (e) AROME$_{analysis}$. Black lines indicate data density. This figure only uses data from the test set.

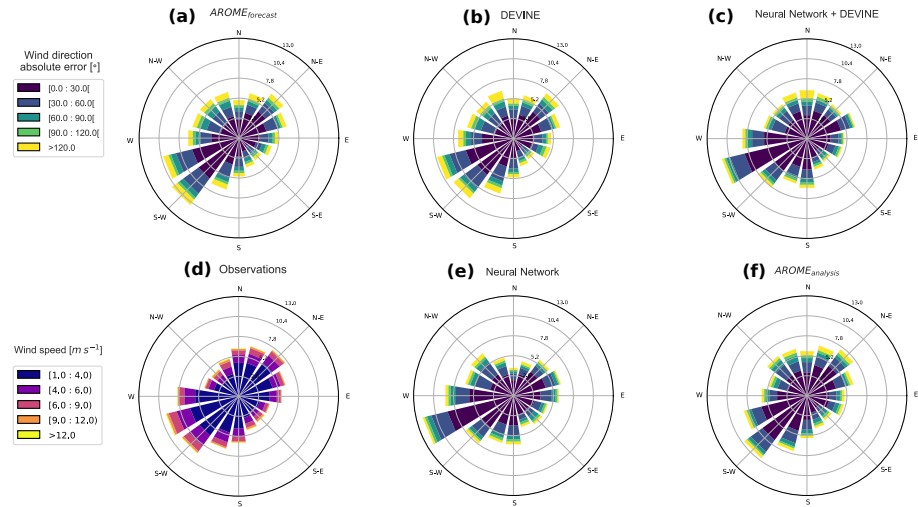

**Figure 5.** Wind roses of modeled wind directions for (a) AROME$_{forecast}$, (b) DEVINE, (c) Neural Network + DEVINE, (e) Neural Network, (f) AROME$_{analysis}$ and (d) observed wind directions. Colors in (a), (b), (c), (e), and (f) indicate wind direction modeling error, obtained by comparing modeled to observed wind directions. Colors in (d) indicates the speed category of the observed wind. Only wind directions acquired for observed and modeled wind speeds above $1\,m\,s^{-1}$ have been considered. Numbers on the radial axis indicate the proportion in % of data in each bin compared to the whole dataset. This figure only uses data from the test set.


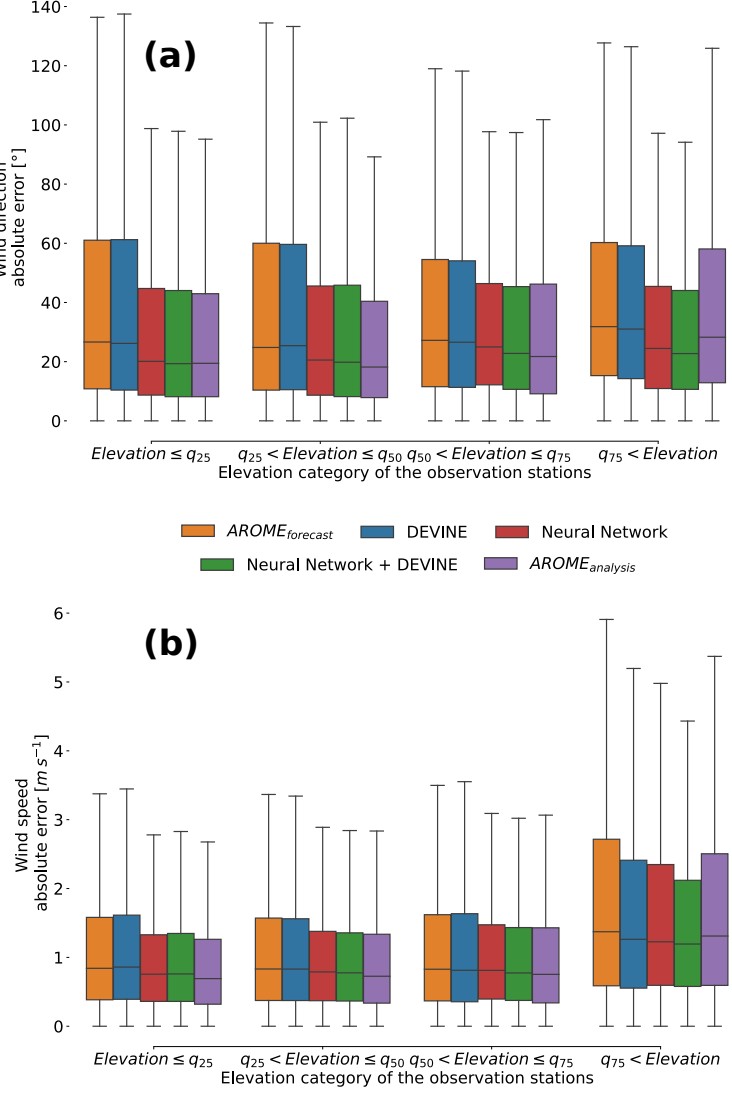

**Figure 6.** Wind direction absolute error (a) and wind speed absolute error (b) categorized by the elevation of the observation station where the measurements were held. In details, the four categories correspond to the four quartiles of the elevation distribution among observation stations: elevation increases from left to right. Each boxplot color indicates a different model. This figure only uses data from the test set.

### 4.3   Influence of forecast lead time and seasonality

In this section, we analyse model performances with respect to forecast lead times (Fig. 7 (a) and (c)) and month of the year (Fig. 7 (b) and (d)). We note that in our study, a forecast lead time has a one-to-one relationship with the hour of the day. In terms of speed, $AROME_{forecast}$ errors are characterized by a peak occurring for lead times between 10h and 20h i.e. mostly during midday and afternoons, in phase with the daily peak of average wind speed. This peak vanishes with $AROME_{analysis}$ which





shows considerable improvements compared to the forecasts. Moreover, we observe that DEVINE evidences small, yet notable

improvements compared to AROME$_{forecast}$. Neural Network shows general improvements compared to AROME$_{forecast}$ and DEVINE by shifting down the error curve, but still conserves a peak around lead time 15h. Finally, the use of DEVINE after Neural Network (Neural Network+DEVINE) again diminishes the mean error, in a quite similar manner to the use of DEVINE after AROME$_{forecast}$. Ultimately, we observe that mean errors are lower with Neural Network+DEVINE than with AROME$_{analysis}$ for the largest ($> 18$) and lowest ($< 8$) lead times.

We obtain similar model rankings in terms of wind direction. Nevertheless, we observe that AROME$_{forecast}$ direction error is marked by a minimum around 12h, which is interestingly shifted from the maximum in speed error observed at 15h. This minimum is shifted by one hour and intensified in AROME$_{analysis}$ and not modified in Neural Network nor in Neural Network+DEVINE. The modification added by DEVINE to the evaluation metrics are low in terms of direction. However, a clear diminution of the error is observed when using Neural Network and Neural Network+DEVINE, which underlines the

added value Neural Network, much over DEVINE, in terms of directional predictions. Similarly to speed predictions, the best statistics among all products are obtained with Neural Network+DEVINE over the largest part of the day, distinctively for the lowermost and uppermost lead times when it outperforms AROME$_{analysis}$.

When modeling errors are interpreted with regards to the month of the year, we observe a peak in speed error during winter months (Fig. 7, (b) and (d)). This observation is consistent with the fact that in mountainous terrain highest wind speeds often

occur in winter (Kruyt et al., 2017). Model intercomparison highlights a similar ordering between models to what happens at the daily scale. The use of Neural Network notably shifts down the error curve. Ultimately, Neural Network+DEVINE compares well with AROME$_{analysis}$ notably during winter months when it outperforms it. Oppositely to wind speeds, wind direction errors do not evidence any dependence on seasonality. Model ordering is however comparable to the ordering concerning speed metrics, with the difference that the use of DEVINE does not evidence any improvements in terms of aggregated metrics. Again,

Neural Network+DEVINE permits to reduce wind modeling error, with a reduction leading to lower errors in winter compared to AROME$_{analysis}$.

### 4.4   Influence of the loss function

The design of an appropriate loss function was important to ultimately obtain the best performing model, presented in this study. The function used to optimize ANN$_{speed}$ ($\mathcal{L}_{speed}$) permits to obtain better integrated metrics (MAE, RMSE and Pearson corre-

lation coefficient) but also to capture wind speed distribution closer to the observed speed distribution. As demonstrated in Fig. 8 which compares observed speed quantiles to simulated quantiles, the use of $\mathcal{L}_{speed}$ shortens the gap between AROME$_{forecast}$ quantiles and the 1-1 line. When fitting the ANN$_{speed}$ with a classical MSE loss function, we obtain a speed distribution with Neural Network+DEVINE which overestimate low quantiles and underestimates high quantiles, i.e. has a tendency to squeeze results around a mean value as already observed by (Dujardin and Lehning, 2022) for similar applications. The improvements

observed after using $\mathcal{L}_{speed}$ are most notable for high wind speed, which is consistent with the different terms composing $\mathcal{L}_{speed}$ (see Sect. 3.3.3). This however contrasts with a degradation of the simulation of very low wind speeds: emphasizing the correction of high wind speeds comes with the cost of putting less penalty on lower wind speeds, and hence results in a

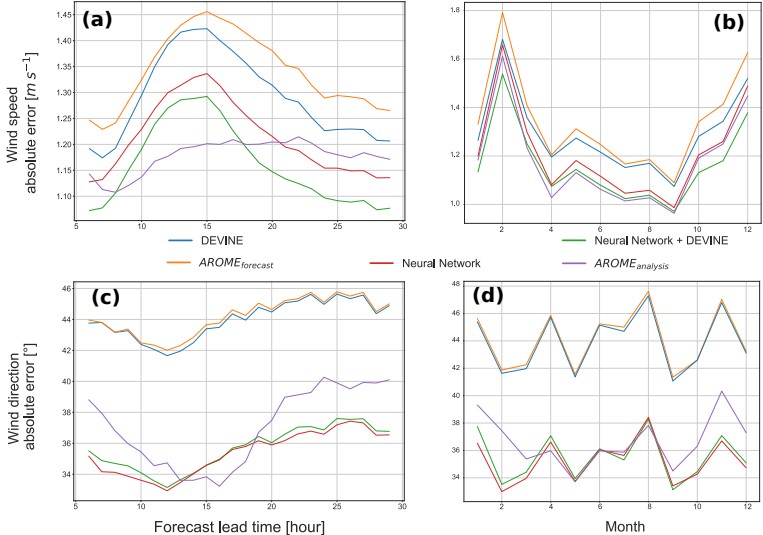

**Figure 7.** Wind speed absolute error as a function of the forecast lead time (a) and month of the year (b). Wind direction absolute error as a function of the forecast lead time (c) and month of the year (d). Color indicate the different models. This figure only uses data from the test set.

model that performs worse concerning first speed quantiles. The use of MSE in place of $\mathcal{L}_{speed}$ to optimize $\mathrm{ANN}_{speed}$ also deteriorates integrated metrics, illustrated by a 12% increase in MAE on the test set. We did not design a custom loss function

for direction but simply selected $\mathcal{L}_{direction}$ (Eq. 2) which immediately yielded satisfactory results.

### 4.5    Sensitivity to the geographical situation

When fitted using observation from the Alps, Neural Network+DEVINE yields poor evaluation metrics in terms of speed when evaluated against data from other mountain ranges, but performs well for downscaling wind direction. We evaluate the ability of our models to correct and downscale $\mathrm{AROME}_{forecast}$ over 18 AWS in Corsica and 21 AWS in the Pyrenees mountain

ranges, which are all located hundreds of kilometers from the Alps, and exposed to different weather regimes. Data from these ranges were not used during training. In Corsica and Pyrenees, Neural Network+DEVINE systematically degrades RMSE, MAE and Pearson correlation coefficient for wind speed when compared to $\mathrm{AROME}_{forecast}$ and $\mathrm{AROME}_{analysis}$ (Table 4). As an illustration, the RMSE increases by 7% with Neural Network+DEVINE compared to $\mathrm{AROME}_{forecast}$. Oppositely, we observe that DEVINE alone improves $\mathrm{AROME}_{forecast}$ metrics in a similar manner as in the evaluation performed in the Alps

(Table 3). Surprisingly, the evaluation of wind direction highlights improvement with Neural Network+DEVINE with respect to $\mathrm{AROME}_{forecast}$ (MAE is reduced by 6°), whereas DEVINE is again not influencing mean wind direction. Wind direction from Neural Network+DEVINE are however on average less precise than with $\mathrm{AROME}_{analysis}$, contrary to the Alpine situation. We can hypothesize that since $\mathrm{ANN}_{direction}$ input variables include almost only variables of topographic nature, correction added by $\mathrm{ANN}_{direction}$ are more linked to local topography than to meteorological situations and hence better generalize to other

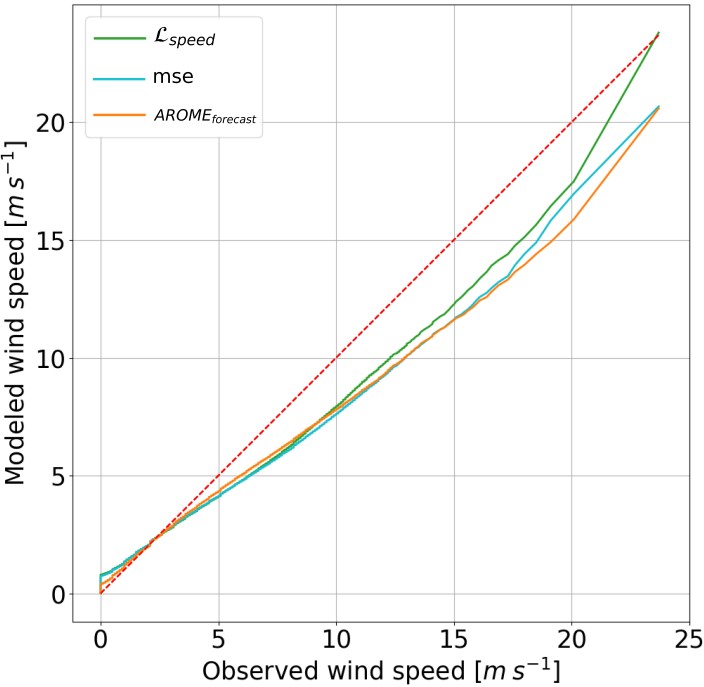

**Figure 8.** Plot of the observed quantiles versus the modeled quantiles for different models. A perfect simulation would present all quantiles along the 1-1 line (red). Each color refers to a single model. mse refers to Neural Network+DEVINE optimized using a mean square error loss function and $\mathcal{L}_{speed}$ to the reference simulation i.e. Neural Network+DEVINE optimized using $\mathcal{L}_{speed}$ (Eq. 3). This figure only uses data from the test set.

mountain ranges. This exploration of the extrapolation abilities of our models to other mountain ranges points towards the need of additional trainings if the models are to target areas outside the Western (french and Swiss) Alps. It however confirms the generic character of DEVINE as highlighted already in Le Toumelin et al. (2022), that does not require any further calibration to be applied to a diversity of Alpine-type mountain ranges.

### 4.6 Case study

To illustrate the added value of Neural Network+DEVINE compared to DEVINE alone, we selected a case study at a mountain observation station located near Piz Corvatsch in South-Western Switzerland (latitude=46.41, longitude=9.82, elevation=3294m). On the 8th of October 2019 at 06h00 (UTC), $AROME_{forecast}$ simulates calm wind conditions ($1\ m\,s^{-1}$) for a wind coming from the South-West (242°). DEVINE downscales the large scale wind field of $AROME_{forecast}$ to a local scale. As a result, it increases $AROME_{forecast}$ wind speed up to $1.47\ m\,s^{-1}$ at the close vicinity of the location of the AWS, since

the site is localised on a ridge, prone to wind acceleration (Fig. 9). On the contrary to both $AROME_{forecast}$ and DEVINE, the observation indicates a wind coming from the North-West (329°) and a much larger speed ($6.4\ m\,s^{-1}$), which is also





**Table 4.** Evaluation metrics obtained by comparing simulation to observed data in other mountain ranges (Corsica and Pyrenees) than the one used during training (Alps). No data from Corsica (18 AWS) nor Pyrenees (21 AWS) was used during training. MAE designates the mean absolute error, RMSE the root mean square error and $\rho$ the Pearson correlation coefficient. The mean absolute error for wind direction was computed by taking care of the cyclic nature of wind direction.

| Variable | Metric | AROME$_{forecast}$ | DEVINE | Neural Network | Neural Network+DEVINE | AROME$_{analysis}$ |
|---|---|---|---|---|---|---|
| Speed | MAE [$m\,s^{-1}$] | 1.53 | 1.51 | 1.69 | 1.64 | **1.23** |
| | RMSE [$m\,s^{-1}$] | 2.28 | 2.20 | 2.49 | 2.38 | **1.85** |
| | Mean bias [$m\,s^{-1}$] | -0.37 | **-0.13** | -0.52 | -0.28 | -0.31 |
| | $\rho$ [] | 0.71 | 0.73 | 0.65 | 0.67 | **0.82** |
| Direction | MAE [°] | 40 | 40 | 34 | 34 | **30** |

partially captured by AROME$_{analysis}$ which indicate a direction of 293° and a speed of 1.81 $m\,s^{-1}$. This example sheds light on high discrepancies than can affect DEVINE input variables (5.4 $m\,s^{-1}$ speed error, 87° direction error). Oppositely, Neural Network modifies AROME$_{forecast}$ wind direction by introducing a 80° clockwise direction change, which puts the

direction closer to the observations. Similarly, Neural Network multiplies the speed by a factor 2.6 ultimately reaching a value of 2.7 $m\,s^{-1}$. After Neural Network, DEVINE downscales these modified large scale conditions. As typically observed with DEVINE, modification in wind directions are modest. However, the speed reaches 3.02 $m\,s^{-1}$, reducing the initial error by 31%. Since the optimization of Neural Network has been obtained after backpropagating error gradients through DEVINE and both ANNs, we can expect that the deep learning model is aware of the expected effect of DEVINE and prevents from

overcorrecting AROME$_{forecast}$. By scrutinizing the day before and after this specific meteorological situation, we observe that AROME$_{forecast}$ systematically underestimated wind speed at this specific location, which is partially corrected by Neural Network+DEVINE. However, this model chain is also responsible for lowering the speed temporal variability, which was already too low with AROME$_{forecast}$. During this period, the direction shifts from North-East to a West direction. Largest modeling errors are observed during the transition period, where Neural Network+DEVINE contributes to bridge the gaps to

observations. During the last hours, AROME$_{forecast}$ captures a more correct wind direction at the station, and the added value of Neural Network+DEVINE is lower. Neural Network+DEVINE however still keeps its ability to spatialize the wind signal over the study area, which is necessary for many application that require high resolution forcings in complex terrain.

## 5 Discussion

### 5.1 Performances and modularity of the chosen architecture

Neural Network+DEVINE evidences improved metrics when compared to AROME$_{forecast}$ and DEVINE both in terms of speed and direction. This is highlighted by more accurate 1-1 plots for wind speed (Fig. 4), better wind distributions (Fig. 5),

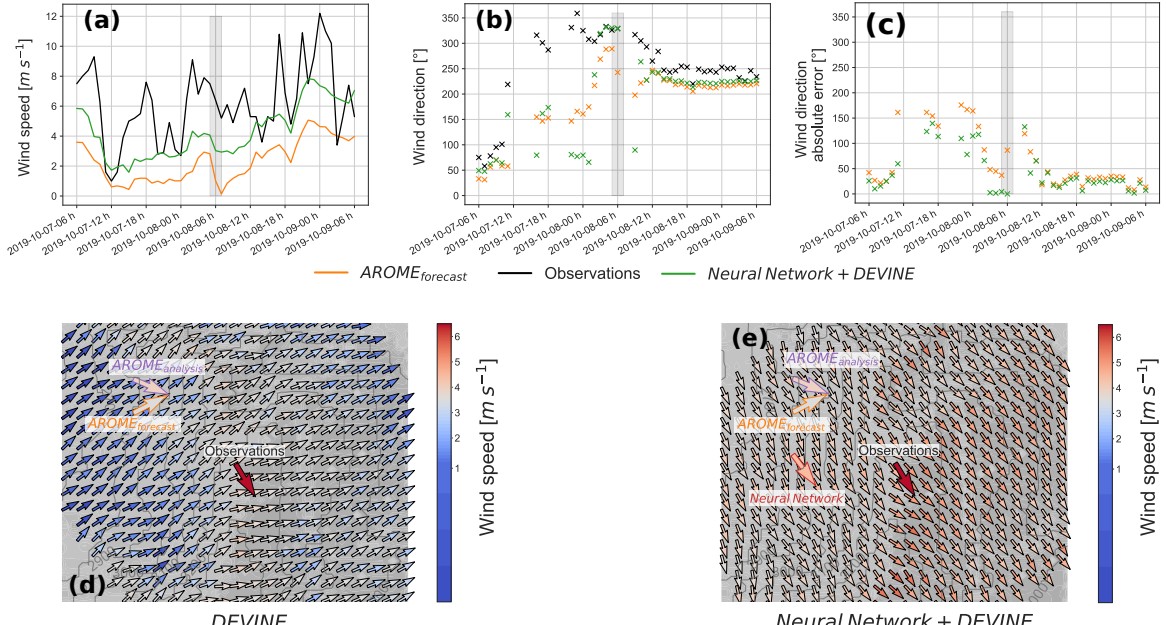

**Figure 9.** Use case of Neural Network+DEVINE at Piz Corvatsch in Switzerland for the 7th to the 9th of October 2019. (a) presents a time series of observed and simulated wind speeds, (b) presents observed and simulated wind directions and (c) modeling errors in wind direction. (d) represents a 2D view of the wind map around the station for the 8th of October 2019 at 06h00 UTC. This date corresponds to the shaded areas in (a), (b) and (c). Small arrows were obtained using DEVINE, fed by $AROME_{forecast}$. (e) was similarly obtained but using Neural Network+DEVINE. $AROME_{analysis}$ and Neural Network (intermediate result) are shown for interpretation. Colors inside arrows indicate wind speed: red colors indicate speed larger than $AROME_{forecast}$ and blue arrows the opposite. The geographical position of $AROME_{forecast}$ corresponds to the location of the nearest grid cell from the observation station in the AROME grid. $AROME_{analysis}$ and Neural Network are located at the same exact position but were moved in the close vicinity of $AROME_{forecast}$ for visual purposes. The high resolution arrows in (d) and (e) correspond to the downscaled signals by DEVINE and Neural Network + DEVINE, characterized by a lower grid spacing than the other wind products ; these arrows were initially distant from 30m and have been downsampled to 90m for visual purposes. This location (Piz Corvatsch) and times (7th to the 9th of October 2019) are only found in the test set.

lower speed and direction errors when errors are categorized by elevation (Fig. 6), forecast lead time or month (Fig. 7) and improvements in integrated metrics (RMSE, MAE and correlation coefficient, Table 3). Evaluation metrics obtained with Neural Network+DEVINE sometimes overpass metrics obtained with $AROME_{analysis}$, for example at elevated stations or during

winter month, suggesting that our method induces notable added value when compared to other well known atmospheric products. Even though comparisons between Neural Network+DEVINE and $AROME_{analysis}$ are limited by a scale discrepancy, supplementary analysis shows that the comparison still holds when $AROME_{analysis}$ is downscaled to a 30m horizontal grid spacing with DEVINE (Fig. S1 in supplementary material). Improved evaluation metrics are all the more encouraging as metrics have been obtained using a spatio-temporal extrapolation assessment, i.e. testing the model at locations not included in the

training set and for a year not included either. This corresponds to a very strict evaluation procedure which raises significant





challenges compared to a simpler evaluation procedure only performed at the sites included in the training set (Bolibar et al., 2020; Dujardin and Lehning, 2022).

The modular architecture of Neural Network+DEVINE appears to us as one of its greatest assets. Decoupling the spatial interpolation of wind fields (in DEVINE) from its correction (in Neural Network) makes the model robust to new NWP systems or NWP version evolutions. Indeed, if a new version of AROME$_{forecast}$ was to be released with important changes, possibly breaking the learned relationships between input variables and observed wind, our architecture permits to simply bypass Neural Network and rely on DEVINE while a new fit is performed with the new NWP version. The same reflection applies to the use of Neural Network+DEVINE at other mountain ranges. As demonstrated in Sect. 4.5, the full model chain is not directly operable on mountain ranges where no data was used during training. As a consequence, a training step is required to adapt Neural Network weights and biases to learn geographically variable relationships between inputs and labels (mostly for ANN$_{speed}$). In the meantime, and contrarily to more classical models that perform model output statistics, the user could rely on the standalone use of DEVINE which showed good generalization capabilities to other alpine-type mountain ranges. Reversely, if user applications that require high resolution wind forcing are not only dependent on the spatial structure of the signal but also require a high degree of plausibility of the downscaled values, the integration of a training phase in the pipeline is possible and would lead to an optimized version of the downscaling scheme. This flexibility doesn't exist in downscaling methods that do not incorporate any fit to observed data.

Since we didn't modify the DEVINE downscaling model in this study, but only added upstream modifications related to coarse scale wind fields, our new architecture inherits the pro and cons of the downscaling model when it concerns the local structure of simulated wind fields. On the one side, using DEVINE favors the simulation of spatially consistent three dimensional outputs at a local scale since DEVINE was built to replicate the structure of outputs provided by an atmospheric model (Helbig et al., 2017). On the other side, DEVINE limitations persist, which is illustrated for instance by the absence of local scale turbulent structure in the wind outputs (Le Toumelin et al., 2022).

In addition to potential applications in wildfire spread modeling, wind energy forecast, wind energy potential assessment, pollutant dispersion evaluation, drifting-snow modeling and avalanche hazard forecasting (Giovannini et al., 2020; Wagenbrenner et al., 2016; Dujardin and Lehning, 2022; Lehning and Fierz, 2008), other applications are sensitive to the accuracy of wind forcing in mountainous terrain. For instance, meteorological forecasters rely on accurate wind predictions in mountains for weather nowcasting and short-term forecasting: they could benefit from the use of a high resolution product such as Neural Network+DEVINE since the modeling chain yields improved wind values when compared to other products (e.g. AROME$_{forecast}$ and AROME$_{analysis}$) under specific topographic and weather situations. Other examples are the use of physics-based models for research purposes on past and future trends in water availability, glaciers evolution and more generally environmental changes. These models often require meteorological information such as wind speed at various scales of interest including the hectometric scale. For instance Réveillet et al. (2018) showed the importance of correctly simulating wind speed in order to simulate the mass-balance of a medium-sized Alpine glacier when using an energy-balance model, an issue that concerns past simulations as much as future projections. Since input variables used in Neural Network are standard NWP output and topographic indicators derivable from DEMs, we hypothesize that Neural Network could be trained by using





reanalyses (e.g. SAFRAN, Vernay et al. (2022), ERA5 Hersbach et al. (2020)). On top of a capability to downscale reanalysis wind fields in the past, this could enable to also downscale the wind of climate projections bias-corrected against these reanalyses, at the instance of the ADAMONT projections (**?**) widely used in France.

### 5.2 Neural network explainability

$ANN_{speed}$ input features have various, unequal and non-linear contributions to Neural Network+DEVINE outputs (Fig. 10), as estimated using ALE (see Sect. 3.5.2). In summary, ALE determine the effect of each input variable on the average output, conditionally to values of an input feature. Most prominent effects are observed for $Wind\,Speed_{10m}$ input feature (Fig. 10, (p)), which confirms our expectations since it corresponds to the downscaled variable, modified in $ANN_{speed}$ output. The use of a skip connection (Fig. 2) for this variable may also play a role in maintaining it as the most important variable for downscaling

even though correlated variables, such as wind speed at other atmospheric levels, are also used as inputs.

Wind speed computed by $AROME_{forecast}$ at other atmospheric levels evidence strong effects on the outputs, most notably when it concerns high speed values. Topographic parameters also have strong impacts on the speed outputs, particularly when it concerns the tails of the parameter distributions. Real elevation ($elevation$, Fig. 10 (g)) and model elevation ($elevation_{model}$, Fig. 10 (f)) have opposite effects: the first one tends to be positively correlated to speed outputs, the second one presents a negative correlation. Two dimensional ALE plots (not shown) suggest almost no second order interaction between both

variables. The joint effect of these variables, approximated by the sum of the first and second order effects, suggests increasing speed outputs with increasing elevation. This confirms our initial interpretation of $AROME_{forecast}$ biases (see Sect. 4.1) that highlighted an average underestimation of speed by $AROME_{forecast}$ over elevated regions. Interestingly, the $TPI_{500m}$, which was also a variable we identified to possibly account for $AROME_{forecast}$ biases, presents diverse effects on the outputs. As

Neural Network+DEVINE was trained after comparing observed values to downscaled simulations, the effects in Fig. 10 do not only compensate for biases in $AROME_{forecast}$ but can also relate to local scale effects in the downscaling module, i.e. counterbalancing missing or incorrectly represented local processes in DEVINE. Finally, we observe that input variables related to the state of the atmosphere (green shaded areas in Fig. 10) have a lower influence on the output and tend to be less dispersed. Interestingly, we see that net shortwave radiations at the surface ($SW_{net}$, Fig. 10 (c)) increase the speed outputs. This

supports Le Toumelin et al. (2022) who observed speed underestimation with $AROME_{forecast}$ during afternoon of summer month, where $SW_{net}$ are generally high. On the opposite, net longwave radiations ($LW_{net}$, Fig. 10 (a)), 2-m temperature ($T_{2m}$, Fig. 10 (b)), cloud cover (Fig. 10 (e)) and the local slope (fig. 10 (j)) show very modest influence on the outputs. Removing iteratively slope and cloud cover from the input features, which are the less impactful input variables according to ALE, and re-training the model, did not impact the evaluation metrics. However, removing all variables with a low ALE ($LW_{net}$, $T_{2m}$,

cloud cover and slope) starts to evidence modifications in evaluation metrics, with for instance the correlation coefficient dropping from 0.72 to 0.70. This could be due to (i) feature interactions not observed in one dimensional ALE plots, (ii) some unexpected over-fitting of the test set, and (iii) visualization artefact from Fig. 10. Indeed, Fig. 10 highlights the largest effects on the outputs, making ALE close to $1\ m\,s^{-1}$ look negligible, however we remind that $1\ m\,s^{-1}$ almost accounts for 50% of the mean speed value ($2.23\ m\,s^{-1}$ in $AROME_{forecast}$). ALE appear here as useful for model interpretation, but also as



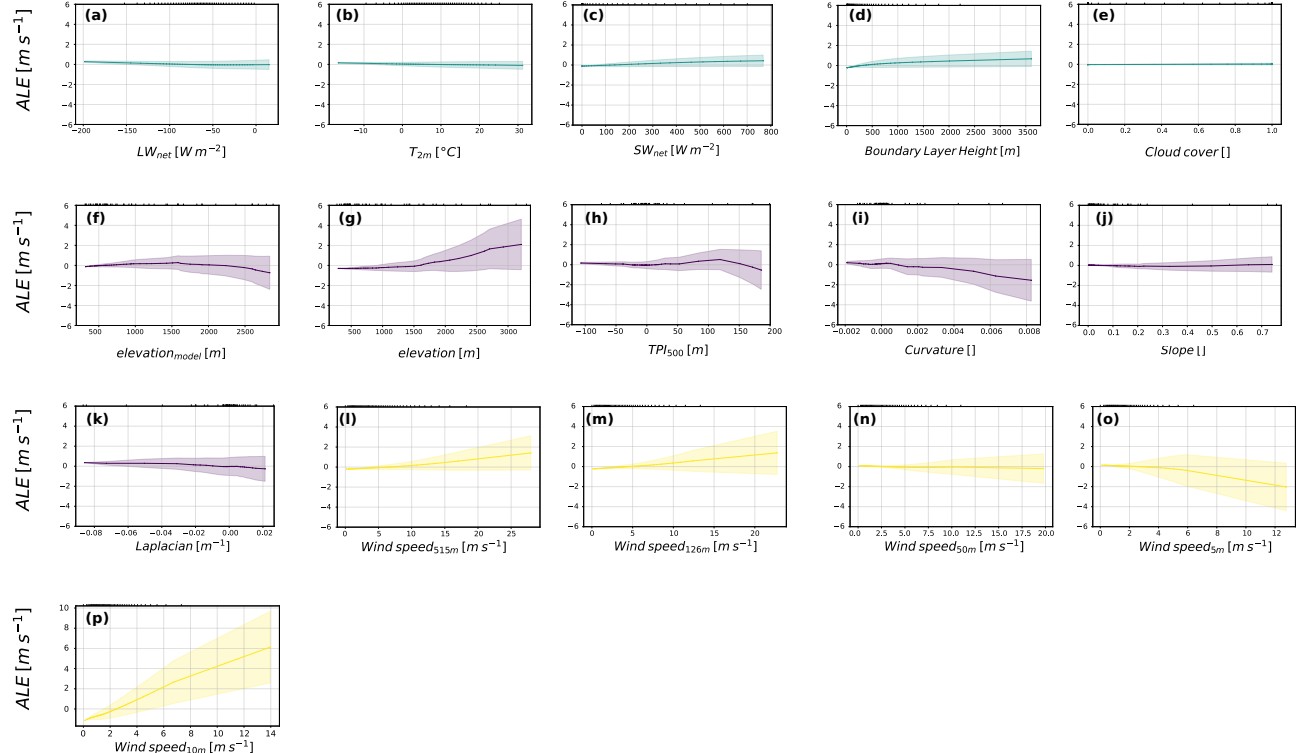

**Figure 10.** Accumulated Local Effect (ALE) associated with each input variable of $\text{ANN}_{speed}$. ALE are expressed in $m\,s^{-1}$ and can be interpreted as the effect of a specific variable at a certain value compared to an average prediction. Shaded areas indicate cumulative dispersion around the mean local effects and increase in each panel from left to right by design. Variation in shaded areas indicate additional uncertainty and comparisons of shaded areas accross panels give insights on uncertainty differences in ALE computation. The small ticks at the top of each panel represent distribution quantiles that were used to compute ALE. Green, violet and yellow colors indicate respectively input meteorological variables, topographic variables and wind related variables.

a tool for input variable selection. Indeed, we can distinct three groups of input features with unequal importance within the model (topographic variables, wind related variables and other weather related variables). This is partly supported by additional sensitivity tests that reveal a larger increase of the RMSE when removing the topographic variables (RMSE=1.68 $m\,s^{-1}$ versus 1.62 $m\,s^{-1}$ with all variables included), or the wind related variables with the exception of $Wind\,speed_{10m}$ (RMSE=1.67 $m\,s^{-1}$) from $\text{ANN}_{speed}$ input features, than when removing other meteorological variables (RMSE=1.65 $m\,s^{-1}$). This is of 555 interest for the application of the Neural Network + DEVINE correction and downscaling strategy to a variety of products like reanalyses, as solely topographic, or topographic plus basic atmospheric variable may be easier to access, retrieve and process, than a complex suit of ancillary weather variables not always available in the reanalysis archives.

Input variables of $\text{ANN}_{direction}$ present scattered individual effects, probably evidencing large interactions among input variables as visible on PDP (Fig. 11). Before using PDP for $\text{ANN}_{direction}$, we asserted that input variables were not correlated




one to each other by checking Pearson correlation coefficients. We studied the impact of input features on the directional difference added to NWP direction on the final on $\text{ANN}_{direction}$ output neuron rather than on the value of the downscaled wind direction, as a classic mean value is not defined for cyclic variables such as wind direction. Note that the modifications computed by $\text{ANN}_{direction}$ also take into account DEVINE effects, which is modest concerning wind direction, and can eventually influence model interpretation. The $Wind\,Speed_{10m}$ does not modify the mean direction, which was expected.

The mean effect for wind direction fluctuates around 0 which suggests some small adjustment given certain azimuths. The mean effect of aspect is also close to 0, excepted around 50 and 100°. The three aforementioned variables were not expected to have a mean effect, which is arguably confirmed by the PDP. However interactions among variables could be anticipated, which is also suggested by the large dispersions around mean effects. Contrarily, $\alpha$ has a strong effect for negative values, which was intuitively expected since this variable already incorporates some interaction between wind direction and aspect.

Surprisingly we do not observe the same behavior for positive $\alpha$. We remind that highest absolute values for $\alpha$ are obtained when a flow arrives perpendicular to a steep vertical slope. The large dispersion around each PDP mean value suggests different scenarios and large variable interactions. Finally, we underline the fact that interpretability methods are not only important to understand how a model deals with inputs and for feature selection, but also to anticipate model output modifications linked with future evolutions of the model providing input data (here $AROME_{forecast}$). As discussed in the previous section, NWP are

under constant evolution, frequently incorporating new or modified parameterizations that tend to modify the model's general behavior and affect several atmospheric variables. Interpretability methods such as in Fig. 10 permit to approximate typical effects that can be obtained through the correction and downscaling model, and anticipate the upcoming of possible modeling errors following NWP updates.

## 6 Conclusion and perspectives

Understanding the complex patterns that characterize wind in mountainous terrain is of high importance for several applications, with direct consequences on environment and human societies. Despite years of continuous improvements, NWP models still rely on downscaling techniques to represent wind features at a local scale in mountains. Not only do the typical km-scale spatial resolution limits their use for several applications, but NWP models are also affected by systematic errors linked to typical meteorological or topographic situations. In this study, we used a large network of observation stations to identify and

apprehend $AROME_{forecast}$ systematic errors. We observed a strong link between model biases and topographic parameters (joint effect of elevation and $TPI_{500m}$) but also a tendency to underestimate largest observed speeds.

Aware of the aforementioned limits, we here designed a new post-processing architecture, called Neural Network+DEVINE, with both the purposes of correcting $AROME_{forecast}$ errors (i.e. applying model output statistics) and increasing the spatial resolution of the wind signal (i.e. downscaling). This new combined architecture (i) benefits from the use of two artificial neural

networks to sequentially correct the coarse scale wind signal for direction and speed according to specific meteorological and topographic situations, before (ii) using the statistical downscaling model DEVINE for the spatial interpolation of the wind fields.

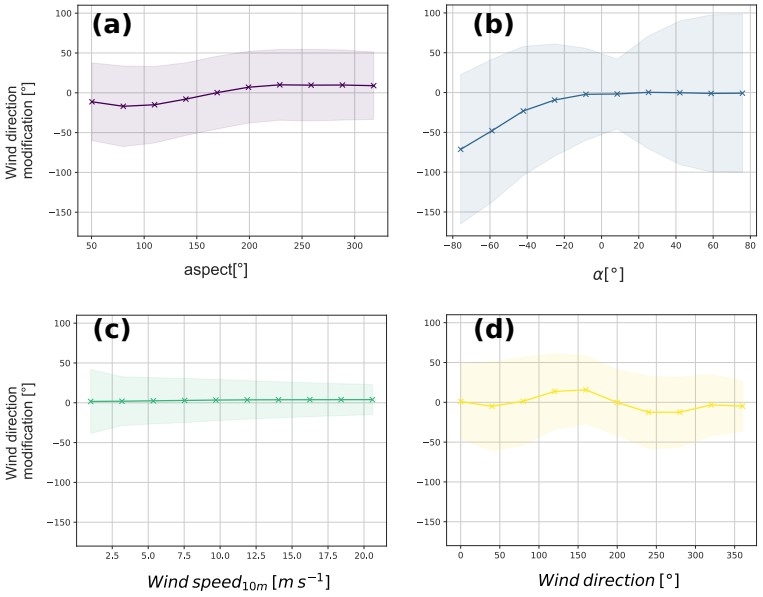

**Figure 11.** Partial Dependence Plot (PDP) for each input variable of ANN$_{direction}$. PDP represent the mean prediction value obtained after fixing all instance of a specific input variable to a certain value. Shaded areas indicate one standard deviation around the mean effect.

This hybrid architecture yields better integrated metrics (MAE, RMSE, mean bias and correlation coefficient) compared to previous alternatives. The evaluation metrics show performances similar to $AROME_{analysis}$, a system benefiting from

assimilation techniques to estimate the most plausible state of the atmosphere in complex terrain. Notably, the best results are obtained at elevated and exposed stations, and also during winter months and more generally for simulating the largest observed speeds, which suggests our new method is well tailored for drifting-snow applications.

  This new type of downscaling model greatly benefits from its modular architecture on several points. By making a distinction between correction and downscaling, our design adds flexibility to the different use cases of our model: it is now easy to either

use the optimized version (Neural Network + DEVINE), or only rely on DEVINE downscaling models when required. Finally, the whole architecture permits to output consistent three dimensional wind fields previously corrected with wind observations. This is a direct consequence of relying on DEVINE for modeling winds at a local scale, an advantage that is counterbalanced by the fact that DEVINE limitations are also inherited by our new architecture.

  This work also stresses the potential of deep learning techniques for the correction of other near-surface atmospheric vari-

ables. The general architecture designed here, with a model tailored to correct large scale errors followed by a more general downscaling scheme could favourably be applied for the bias-correction and downscaling of other variables like 2m air temperature, that similarly exhibits high spatial variations in complex terrain in relation with topographic and meteorological gradients.

  Future work should include a generalization of our model to other forecast cycles. Indeed, we only used here forecasts

initialized from the 00h00 analysis, making our model a proof of concept that needs to be generalized to other forecast cycles.





Furthermore, our design adds up to a large array of existing solutions to downscale wind fields in complex terrain for which an intercomparison project is highly required. Such a project could include the use of dense observational networks to assess precisely the behavior of wind at a local scale. This exercise could help listing the pro and cons of each methods, often developed over different areas and targeting distinct end user application cases, and reveal each method value for operational applications. The wealth of near-surface observations to be acquired at high spatial resolution in the central European Alps within the TeamX campaign (Serafin et al., 2020), complemented with the observations routinely acquired by the local met-services, will provide an adequate database for this venture.

*Code availability.* The code used to build, train and evaluate the model is available on https://github.com/louisletoumelin/neural_network_and_devine/.

*Author contributions.* LLT worked on conceptualization, led the investigations, built, trained and evaluated the models, wrote the first draft and designed the figures. IG worked on conceptualization, supervision, investigation and helped to write the first draft. CG and NH worked on conceptualization, provided guidance in scientific developments and helped to write the first draft.

*Competing interests.* The authors declare that they have no conflict of interest.

*Acknowledgements.* This research is supported by the french Meteorological Institute (Météo-France). The authors thank the national observation service GLACIOCLIM (CNRS-INSU, OSUG, IRD, INRAE, IPEV) for the data provided. The authors thank MeteoSwiss, the Swiss Federal Office of Meteorology and Climatology for the services provided.



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
