# Peer review of "A two-folds deep learning strategy to correct and downscale winds over mountains."

_Nonlinear Processes in Geophysics, 2023_

## Author Comment (AC1)

**Reviewer 1**

This manuscript presents a deep learning-based statistical downscaling model for surface winds over complex terrain. The model consists of two parts, correction of winds from a regional atmospheric model, and conversion from a coarser grid to a finer grid, based on information of high-resolution topography data and atmospheric conditions. Results indicate that the proposed model better represent winds over Western Alps, for which the model is trained. The manuscript is generally well written, and conclusions are clear. In particular, analyses on the explainability increases its potential for real applications, where reliability of the model matters. There are, however, still room for improvement in the presentation quality. Therefore, my recommendation is publication after minor revisions are made.

We thank the Reviewer for the time dedicated to the review and for his constructive comments that we believe help improve the manuscript. Please find below our answers to all of your comments.

Minor comments:

1. I understand that the model names, such as AROME and ARPS, are well known in the community, but it would be better to present their full names somewhere in the manuscript, such as the lines at which they appear for the first time or a list of names at the end of manuscript.

   AROME and ARPS acronyms are now fully explained at their first occurrence in the manuscript.

   *As an illustration, Le Tourmelin et al. (2022) observed that AROME (the NWP "Application of Research to Operations at MesoscalE" operated by Meteo-France) wind fields are frequently underestimated at elevated and exposed areas.*

   *More specifically this model was trained at replicating the behavior of the atmospheric model ARPS ("Advanced Regional Prediction System") over complex Gaussian topographies Helbig et al. (2017).*

2. What is the source of high-resolution topography data?

   The digital elevation model used in this manuscript is a combination of DEM from different sources, used for current studies at our research lab (Snow Research Center - CEN/CNRM/CNRS). Within France boundaries, the "RGE ALTI" DEM |[IGN], i.e. a 5m grid-spaced DEM from the national cartography institute, is resampled to 30m and used. Outside of France, the "GLO-30" DEM, i.e. the 30m grid-spaced Copernicus DEM, is used (Fahrland et al. 2020). Both DEM have been previously processed to be assembled in a coherent way.

   The following text has been added:

   *In this study, the DEM used has been obtained after merging RGE Alti DEM resampled to 30m (IGN) inside of France boundaries and GLO-30 DEM in Switzerland (Fahrland et al. 2020).*

3. In this manuscript, the data are divided into training and test datasets. However, in many practices, people divide data into three sets: training, validation, and test, where the validation dataset is used to tune hyperparameters. How did you tune the hyperparameters?

Initially, we started the study using a validation dataset composed of a few stations in order to properly calibrate hyperparameters. However, we found that the results on the validation dataset were highly sensitive to the selected validation stations. This is explained because wind conditions and associated metrics are highly variable in mountainous terrain. Even though we did not stick to this evaluation procedure, this step enabled us to derive a first sketch of the selected architecture.

Another procedure could have consisted in selecting hyperparameters using cross validation. Given the fact that the derivation of the selected architecture required hundreds to thousands of numerical experiences, each experience requiring several hours of computations, multiplying the number of training instances to compute cross validation would not have been possible with the resources at our disposal. These many experiences are explained because we did not use a standard deep learning architecture but built a rather more complex data flow that required much experimentation.

In the end, we kept the hyperparameters both inherited from the first experiments using a validation dataset and from the next experiments using only a test dataset. When the architecture and hyperparameters were fixed, we ensured that we did not overfit our network to the test database by training two alternative models with the same architecture and inputs data on two alternative train sets. For each alternative train set, alternative test sets were also obtained after selecting new test stations with the same selection procedure as described in the article. The average three-fold evaluation is presented below:

| Variable | Metric | $AROME_{forecast}$ | Neural Network | Neural Network+DEVINE | $AROME_{analysis}$ |
|---|---|---|---|---|---|
| Speed | MAE $[m\,s^{-1}]$ | $1.32 \pm 0.03$ | $1.20 \pm 0.05$ | $1.16 \pm 0.06$ | $\mathbf{1.15 \pm 0.04}$ |
| | RMSE $[m\,s^{-1}]$ | $1.87 \pm 0.06$ | $1.72 \pm 0.08$ | $\mathbf{1.64 \pm 0.09}$ | $1.67 \pm 0.07$ |
| | Mean bias $[m\,s^{-1}]$ | $-0.04 \pm 0.07$ | $-0.05 \pm 0.08$ | $\mathbf{0.00 \pm 0.04}$ | $-0.06 \pm 0.06$ |
| | $\rho\,[]$ | $0.60 \pm 0.03$ | $0.66 \pm 0.02$ | $\mathbf{0.70 \pm 0.02}$ | $0.69 \pm 0.02$ |
| Direction | MAE $[°]$ | $44.0 \pm 1.55$ | $36.2 \pm 1.4$ | $\mathbf{35.4 \pm 1.22}$ | $37.1 \pm 0.37$ |

Notations in this Table follow the specifications from Table 3 in the main manuscript.

This choice is summarized in the main manuscript as follows:

*Diverse architectures and hyperparameters were tested in order to converge to the final model. We checked that our model doesn't overfit the test set by computing metrics using a three-folds cross-validation strategy, presented in Table S1 in the supplementary material.*

4. Figure 4 and 5: Labels are too small. Please enlarge them.

The labels are now bigger.

5. Figure 6: In the main text, the bottom panel (Fig. 6b) appears first, and the top panel (Fig. 6a) later. Because this is a bit confusing, I would suggest swapping the panels. In addition, I would suggest citing the figures in the main text as Fig. 6a and Fig. 6b by explicitly indicating the panel ID.

We followed your advice and updated the text and figures accordingly.

6. Figure 7: Please enlarge the labels. The lines are difficult to distinguish. You can use different line types, such as dashed or dotted.

We updated the graphics accordingly.

7. Why does this study start downscaling at the forecast lead time of 6 h? I am curious how these models perform at shorter lead times. In other words, do you assume the same model error in AROME_forecast for the lead times from 6 to 29 hours?

We downloaded modeled data using the same procedure as in Vionnet et al. (2016), Quéno et al. (2016), Le Toumelin et al. (2022) and similarly to Gouttevin et al. 2022. In more detail, we only extracted lead times between +6 and +29h, issued from the 00:00 analysis. This allows us to reconstruct continuous atmospheric forcings over the study period, as typically found in the forcing files of snow models such as CROCUS, that are commonly processed at our lab. The choice of +6h was originally selected in Vionnet et al. (2016) as a way to minimize the effect of the analysis at 00:00 on the simulations. Here, we adopted the same procedure in order to have corrected and downscaled simulations comparable to the previously used set of data. This offers the possibility to update previously used forcing files including AROME data in the most consistent way.

However, this strategy does not assume that errors between lead time +6h and lead time +29h are the same. Input variables used by the model allow the neural network to access information correlated to the time of the day, which in our case also corresponds to a given lead time. This way, the correction can take into account varying performances following different lead times. We tested to include the lead time among the input variables but did not observe any improvements. Constructing an architecture that would correct shorter lead time (<6) or larger lead times (> 29) or other analysis cycles (different from 00:00 UTC) would probably require training different models, or informing the models about the lead times/analysis cycles used.

We added the following modifications to the main manuscript:

*"Firstly, we built a continuous time series by extracting +6 to +29h AROME forecast lead times, initialized with the analysis of 00:00 UTC, as in Quéno et al. (2016); Vionnet et al. (2016); Le Toumelin et al. (2022). This was done as a way to obtain continuous time series, typically used to force snow and surface models as in Quéno et al. (2016); Vionnet et al. (2016); Gouttevin et al. (2023)."*

8. Related to the question above, have you applied "Neural Network+DEVINE" to AROME_analysis? "AROME_analysis+Neural Network+DEVINE" may serve as a good analysis product.

We didn't apply "Neural Network+DEVINE" to "AROME_analysis" because we didn't have more data than used in the test set when it concerns AROME_analysis. With the storage system hosting modeled data at our disposal, it requires months of continuous downloading to obtain years of model outputs. As a consequence we can not perform additional training within a short period of time. However, we fully agree with the reviewer that training our model using AROME_analysis as inputs would be of high interest for the community.

9. Figure 9: The color bars for (d) and (e) have no labels for negative values.

The label is now incorporated.

10. Page 25, line 3: A link to a reference is broken.

The link is now corrected.

11. Section 5.2: In this section, each paragraph looks very long. I would suggest splitting the paragraphs for readability.

We thank the reviewer for this advice and agree that this section was hard to read as a single uniform paragraph. We splitted the long paragraph into multiple paragraphs to improve readability.

12. Figure 10 and 11: The yellow shadings and lines are difficult to read.

We changed the contrast and modified the color to a darker yellow to make the shadings and the line more visible.

13. Figure 10: ALE increases as the wind speed increases at 515, 126, and 10 m. In contrast, ALE is neutral for wind speed at 50 m, and ALE decreases as wind speed at 5 m increases. Do you have any interpretation on this behavior?

By construction, the ANN_speed computes a correction that is added to 10m AROME wind speed. Consequently, we understand that model outputs will increase or decrease in phase with this variable, which is confirmed by the ALE plot.

Then, ALE plots for wind speeds from other atmospheric levels give us insights on how input values relate to the output values. This is informative to understand how decisions are made inside the model but not necessarily why decisions are made or if they relate to physical processes.

We note that AROME wind fields result from numerical computations that account for processes from the free atmosphere down to the surface. It is very probable that errors observed in the simulations of 10m wind fields result from modeling error originating in different parts of the code. For example, inaccurate parametrizations of atmosphere surface interaction can translate to errors in low level wind fields (e.g.

5m wind fields). Similarly, erroneous estimations of higher altitudes wind fields can lead to errors in surface wind estimations. We hypothesize that these types of behavior might be (partly) integrated in the decision making of the model regarding wind variables.

14. Page 26, line 558-: Although this paragraph states that input variables perhaps have large interactions, in the following sentence, it is also stated that input variables are not correlated, which is a bit confusing. Could you clarify this point?

Interactions between two variables define how a specific set of values are combined in order to determine the output of the model. In other words, the variables interact one with each other within the model in order to create the output. On the other hand, correlated variables are variables that can be linearly related. These two concepts are independent.

To illustrate this point, we can hypothesize that a model downscaling wind fields depends on large scale wind direction and atmospheric stability. For a given large scale direction, the model might present various outputs given the air stability: the feature "large scale wind direction" interacts with the feature "air stability" to produce the model output. However, large scale wind directions and air stability can be completely independent (i.e. not correlated): given a fixed large scale wind direction, any type of air stability can be encountered in the input data.

We now specify in the text that the interaction between variables designates interaction within the model:

*Input variables of ANN_direction present scattered individual effects, probably evidencing large interactions among input variables within the model when computing the output, as visible on PDP (Fig. 11).*

15. There are many typos and grammatical errors. Please doublecheck.

We apologize for these errors. We carefully read the article and removed all the observed errors.

References:

Fahrland, E., Jacob, P., Schrader, H., & Kahabka, H. (2020). Copernicus digital elevation model—Product handbook. *Airbus Defence and Space—Intelligence: Potsdam, Germany*.

IGN: RGE ALTI ® Version 2.0, https://geoservices.ign.fr/sites/default/files/2021-07/DC_RGEALTI_2-0.pdf.

Quéno, L., Vionnet V. , Dombrowski-Etchevers I. , Lafaysse M. , Dumont M. , and Karbou F. , 2016: Snowpack modelling in the Pyrenees driven by kilometric-resolution meteorological forecasts. *Cryosphere*, **10**, 1571–1589, doi:10.5194/tc-10-1571-2016.

Vionnet, V., Dombrowski-Etchevers, I., Lafaysse, M., Quéno, L., Seity, Y., & Bazile, E. (2016). Numerical weather forecasts at kilometer scale in the French Alps: Evaluation and application for snowpack modeling. *Journal of Hydrometeorology*, *17*(10), 2591-2614.

Gouttevin, I., Vionnet, V., Seity, Y., Boone, A., Lafaysse, M., Deliot, Y., & Merzisen, H. (2023). To the origin of a wintertime screen-level temperature bias at high altitude in a kilometric NWP model. *Journal of Hydrometeorology*, *24*(1), 53-71.

Le Toumelin, L., Gouttevin, I., Helbig, N., Galiez, C., Roux, M., & Karbou, F. (2023). Emulating the Adaptation of Wind Fields to Complex Terrain with Deep Learning. *Artificial Intelligence for the Earth Systems*, *2*(1), e220034.

---

## Author Comment (AC2)

The paper presents a deep learning-based strategy for downscaling wind fields over mountainous terrain. The innovation of this work lies in the versatility of the approach, achieved by separating the downscaling in two parts: i) Correcting large scale data provided by a NWP model that serves as input for ii) a statistical downscaling model that assumes perfect large scale input data. This decoupling ensures the downscaling model remains independent of the NWP models providing the input data. The paper is well structured, the method is mostly well presented and the results are discussed in a satisfactory manner. I only have minor comments of which many are mere suggestions.

We thank the reviewer for his comments that helped to improve the manuscript. We appreciated the time and the care dedicated to our study. Please find below our answer to your comments.

- In general figures should be made a lot cleared with significantly larger font sizes (in particular for ticks) and thicker lines. Please carefully revise all Figures for readability.

  We followed your advice and revised figures 4, 5, 6, 7, 9, 10 and 11 and did our best to improve readability.

- The wording is sometimes slightly peculiar. I would advice having a native English speaker look over the manuscript.

  We apologize for these errors. We carefully proofread the article and removed all the observed errors.

- P2 L28: NWP are --> NWP models are

  Corrected.

- P2 L34: the literature --> literature

  Corrected.

- P2 L35-37: It is not clear to me how Dupuy et al. (2021) and Goutham et al. (2021) relate to Zamo et al. (2016) and Hoehlein et al. (2020). What is the different way? I suggest to either mention what the difference is, or don't distinguish them so clearly in the text.

The main difference between Dupuy et al. (2021) and Goutham et al. (2021) to Zamo et al. (2016) and Hoehlein et al. (2020) is that the former developed statistical models specifically calibrated to operate at individual locations. Application of their model out of their training domain is not documented since their application targeted dedicated sites. Contrarily, Zamo et al. (2016) and Hoehlein et al. (2020) models produce gridded outputs on a predefined grid. This is different from other methods such as Helbig et al (2017) or the DEVINE model that have been developed to be operated over any geographic region (i.e. not restricted to specific locations as in Dupuy et al. (2021) and Goutham et al. (2021), nor specific grids as in Zamo et al. (2016) and Hoehlein et al. (2020)).

The following modifications have been adopted:

*e.g. Dupuy et al. (2021) and Goutham et al. (2021) develop statistical downscaling methods specifically tailored to operate at specific, individual locations (their calibration sites).*

- P2 L38: Not only these methods --> These methods not only

Corrected.

- P2 L50: the literature --> literature

Corrected.

- P3 L58: incorrect --> imperfect

Corrected.

- P3 L59: imprecise initial condition and errors due to the assimilation procedure may be redundant?

Thanks for noticing. Yes it is, we removed "imprecise initial conditions"

- P3 L59: Perhaps etc. is better than …

We changed … into etc.

- P3 L59-63: These sentences arouse the expectation that the manuscript addresses uncertainty quantification of errors. However, I feel that the quantification of errors and uncertainty of errors is not clearly distinguished throughout the manuscript.

This paragraph underlines the difficulty to attribute the error in the modeling chain: NWP -> downscaling model -> results. Are the results good/bad because of correct/incorrect downscaling or due to correct/incorrect initial conditions? The following paragraph suggests that even though error identification might be hard, error compensation is still possible, notably with the methods developed in this study. To clarify this point, we rephrased the sentence as follows:

*Furthermore, the use of a downscaling model also makes it difficult to determine the origin of the modeling errors: whether the downscaling model accurately/inaccurately simulates local-scale processes or if error compensation between the large-scale forcing and the downscaling model scrambles the evaluation.*

- P3 L65: lower --> smaller

Corrected.

- P3 L68: sufficiently the input wind --> the input wind sufficiently

Corrected.

- P3 L72: of DEVINE --> of the DEVINE

Corrected.

- P3 L72: I don't understand how the effect of downscaling is accounted for.

The effect of downscaling is not accounted for during inference: the NWP data are processed and lead to the derivation of a corrected wind field that is in turn used in the downscaling model. The computation of the correction (ANN_speed and ANN_direction) do not receive any feedback from DEVINE during inference.

However, during training, the back propagation algorithm takes into account DEVINE when modifying the neural networks parameters. This is explained by the chain rule that links the derivative of the loss function to the derivative of the neural network parameters. With plain words, the chain rule computes how changes in the DEVINE outputs affect the loss and then how changes in the parameters of the input neural networks affect DEVINE outputs, and link the variations of these quantities.

We rephrased L72 to:

*Indeed, the correction is made before the downscaling step, but the effect of downscaling is accounted for in the optimization of the neural networks' parameters that are responsible for the correction.*

- P3 L76: Again, I would rather use etc than …

Corrected.

- P4 L103: Why discard forecast lead times 1-6? Aren't these very relevant for various applications? Is it because short term forecasts are good enough and would not benefit as much from a corrective step? Please shortly motivate your choice.

For simplicity, we extracted modeled data using the same procedure as in Vionnet et al. (2016), Quéno et al. (2016) and similarly to Gouttevin et al. 2022, i.e. extracting only lead times between +6 and +29h, issued from the 00:00 analysis. This is done as a way to reconstruct continuous atmospheric forcing over the study period, as typically found in the forcing files of snow models such as CROCUS, that are commonly processed at our lab. The choice of +6h was originally selected in Vionnet et al. (2016) as a way to minimize the effect of the analysis at 00:00 on the AROME simulations and hence avoid a major inhomogeneity around 00:00 induced by data assimilation at 00:00. Here, we kept it to finally obtain corrected and downscaled simulations comparable to the previously used set of data, as a way to offer the possibility to update previously used forcing files in the most consistent way.

However, we fully agree that discarding lead times +1 to +5h prevents us from correcting specific lead times of high interest for other applications. Including these lead times in our study would probably require a change in the deep learning architecture, for example by informing the network about the processed lead time (i.e. adding the lead time to the input variable) since, in this hypothetical new configuration, a single lead time would not be equivalent to a single valid hour (more

than 24 lead time extracted) and lead-times close to the analysis time would likely require different treatment within the model as the assimilation induces a high inhomogeneity with respect to later lead-times. Another solution would be to fit another model with the lead times of interest (e.g. +1 to +24h), but by keeping our methodology based on the extraction of continuous time series.

Finally, your reflection on the potential lower errors found in first lead times is very interesting. We had the same intuition, but results from Figure 7 also indicate that for lead times between +6 and +29h, the influence of lead time on the error is of secondary order when compared to the diurnal cycle of the error. It is probable that this result also holds for lead time shorter than +6h.

We added the following modifications:

*"Firstly, we built a continuous time series by extracting +6 to +29h AROME forecast lead times, initialized with the analysis of 00:00 UTC, as in Quéno et al. (2016); Vionnet et al. (2016); Le Toumelin et al. (2022). This was done as a way to obtain continuous time series, typically used to force snow and surface models as in Quéno et al. (2016); Vionnet et al. (2016); Gouttevin et al. (2023)."*

- P4 L108: I believe "dispose of" is misused throughout the manuscript. It means "getting rid of".

Indeed, we didn't use this verb correctly. We changed "dispose of" L108 by "obtain", L226 by "have" and L282 by "obtain".

- P4 L114: introduce acronym AWS

Corrected.

- P4 L120: I suggest to delete the sentence where you mention you use some stations for training and others for evaluations here. It is well explained later and a bit confusing to state it here.

We agree that this sentence adds redundancy and removed it..

- Caption Figure 1: observation --> observations

Corrected.

- Caption Figure 1: Add (not shown) after "Note that an additions data set"

The text has been added.

- P7 L168: time --> times

Corrected.

- P7 L171: How did you choose the hyperparameters?

The choice of hyperparameters was derived following two distinct steps. We started the study by calibrating hyperparameters using a validation dataset composed of several stations. However, the results on the validation dataset were highly sensitive to the selected validation stations, notably because wind conditions are highly variable between stations in mountainous terrain. Even though we derived a first sketch of the model architecture and hyperparameters with this method, we did not stick with this evaluation procedure for the second step of the study.

In a second step, we worked using only a training step for model training and a validation set for model evaluation. Hyperparameters have been selected to give reasonable results on the test set. Ideally, we should have selected hyperparameters using cross validation, i.e. partitioning training/test dataset several times and then optimizing the averaged results obtained using the different test sets. Given the fact that it required hundreds to thousands of numerical experiences to converge to the final solution presented in this manuscript and that each experience requires several hours of computations, using a cross validation procedure would not have been possible with the resources at our disposal.

In the end, we kept the hyperparameters both inherited from the first experiments using a validation dataset and from the next experiments using only a test dataset. When the architecture and hyperparameters were fixed, we ensured that we did not overfit our network to the test database by training two alternative models with the same architecture and inputs data on two alternative test datasets, i.e. we verified that the presented results are consistent with results obtained using a three-folds cross validation strategy. These alternative test sets were obtained after selecting new test stations with the same selection procedure as described in the article.

The averaged results using three distinct test sets, obtained after training the model on three distinct training sets are reported below:

**Table 6.** Evaluation metrics obtained on the test dataset (Alps). MAE designates the mean absolute error, RMSE the root mean square error and $\rho$ the Pearson correlation coefficient. The mean absolute error for wind direction was computed by taking care of the cyclic nature of wind direction.

| Variable | Metric | AROME$_{forecast}$ | Neural Network | Neural Network+DEVINE | AROME$_{analysis}$ |
|---|---|---|---|---|---|
| Speed | MAE $[m\,s^{-1}]$ | $1.32 \pm 0.03$ | $1.20 \pm 0.05$ | $1.16 \pm 0.06$ | $\mathbf{1.15 \pm 0.04}$ |
| | RMSE $[m\,s^{-1}]$ | $1.87 \pm 0.06$ | $1.72 \pm 0.08$ | $\mathbf{1.64 \pm 0.09}$ | $1.67 \pm 0.07$ |
| | Mean bias $[m\,s^{-1}]$ | $-0.04 \pm 0.07$ | $-0.05 \pm 0.08$ | $\mathbf{0.00 \pm 0.04}$ | $-0.06 \pm 0.06$ |
| | $\rho\,[]$ | $0.60 \pm 0.03$ | $0.66 \pm 0.02$ | $\mathbf{0.70 \pm 0.02}$ | $0.69 \pm 0.02$ |
| Direction | MAE $[°]$ | $44.0 \pm 1.55$ | $36.2 \pm 1.4$ | $\mathbf{35.4 \pm 1.22}$ | $37.1 \pm 0.37$ |

- P7 L185: The term "label" might be confusing to some, as it has not been introduced before. I suggest reducing the number of synonymous terms throughout the manuscript. For example, choose either labels, targets or ground truth and use it consistently.

We removed the mention to "ground truth" and "labels" and now only refer to "targets" for consistency.

- P7 L191-195: I don't think it adds value to add technical information here. This information is relevant for those who are interested in the code and should therefore be provided with the code publication.

This paragraph has been removed.

- P10 L226: I don't understand this sentence. I believe you are trying to explain how you deal with the mismatch between the location of the observation and that of the model grid points, but I don't understand it. Please try to clarify.

Yes, this paragraph describes how it is important to extract grid points from our simulated maps that match with observation sites. The text has been modified to clarify this explanation:

*In order to adapt the weights and biases of the ANNs, we adopted a sequential approach. First, we optimized ANN_direction for wind direction, and then optimized ANN_speed for wind speed. This order is motivated by the fact that an erroneous direction can translate into erroneous high-resolution wind speeds with DEVINE as a result of wrong topography adjustments, whereas the opposite will have less impact. We selected 218 training observation stations in the french Alps measuring wind speed and direction. Additionally, we used data from the nearest grid cell of AROME at each of these stations to take into account large scale atmospheric conditions, and extracted topographic maps around these observation stations to take into account topography. It is important to note that in the end, our model outputs wind field maps whereas observation stations provide information for isolated points in space. In order to optimize the neural networks, we selected a single wind value at the center of each simulated map that is compared to the corresponding target from the observation station. This step required to accurately match the position of the center of the simulated map with the observation station: this was possible by providing input topographies to our model that were already centered on the location of the observation sites.*

- P10 L229: This is related to my confusion on how the effect of downscaling is accounted for (see comment on P3 L72). How exactly is DEVINE involved in the neural network for the corrective step? Also, is the "observed ground truth" observations here?

Following your previous comment on the semantic used in the manuscript, we rephrased "observed ground truth" to "target" (which corresponds to observed wind fields). We refer to our answer on your previous comment when it concerns the optimization procedure.

- P1- L235: Perhaps emphasize again that the motivation for not wanting to create a new downscaling model is versatility.

We followed your suggestion and now underline why modifying DEVINE weights would lead to the obtention of a less versatile downscaling model.

*This choice was made because our goal was to develop an optimization system to be used with DEVINE, rather than fitting DEVINE to AROME wind fields. Modifying DEVINE weights would lead to the creation of a new and less versatile downscaling model (see Sect. 5), that assumes a specific type of input data (here AROME data), with potential limitations in its scope of applicability.*

- P10 L245: to produce squeezed --> to produce a squeezed

Corrected.

- Figure 2: In my opinion contains too much details on DEVINE, both in the caption and the figure. Instead, the neural networks could be highlighted more. For example, I was a little confused on how exactly the skip layers are incorporated. For example, I was confused whether the network outputs a correction of the full wind speed/direction fields.

The effect of ANN_speed on NWP speed is the following :

$$ANN\_speed(topo\ variables,\ nwp\ speed,\ nwp\ dir.) = x\ +\ nwp\ speed \qquad (1)$$

where $x\ =\ neural\ network_{speed}(topo\ variables,\ nwp\ speed,\ nwp\ dir.)$ i.e. x encompasses only the effect of the neural network.

The full output of ANN_speed adds "$x$" to it the original NWP wind speed "$nwp\ speed$" through the skip connection (+ in the 1st equation).

The skip connection helps to compute a corrective term $x$ that is in turn added to the $nwp\ speed$ to form the modified speed used to force DEVINE.

This is represented in Fig. 2 by the connection on the margin of the neural networks but also by the "+" sign representing the operation from Eq. (3).

We simplified the figure caption as follows:

*Scheme of the new model architecture. The architecture is composed of two artificial neural networks (ANN) in addition to DEVINE downscaling model. The first ANN predicts wind direction (orange, ANN_direction), the second one predicts wind speed (blue, ANN_speed). Skip connections are represented by blue and orange connections, associated with the sign "+". Modified wind speed, direction and a high resolution topographic map are then sent to DEVINE Le Toumelin et al. 2022, which in turn outputs maps of the three components of wind fields (U, V, W) at high resolution (30m). During the training step, wind direction and speed values are computed at the maps' center (taken to coincide with an observation station) and sequentially compared to in situ observation using appropriate loss functions (see*

*Sect. 3.3.3). ANN_direction and ANN_speed optimizations are guided by the gradient of these losses. We note that ANN_direction and ANN_speed have both an independent model architecture, including the nature of input variables. "Topo. variables" refer to input variables of topographic nature, "NWP variables" to inputs corresponding to forecasted meteorological and surface variables, "NWP direction" to forecasted wind direction and "NWP speed" to forecasted wind speed. The mathematical operations performed within DEVINE are listed using colored boxes and are explicited in Le Toumelin et al. 2022. Finally, red dots following ANN_direction and ANN_speed consist of the intermediate results of the network (i.e. the ANN outputs) and are referred to as Neural Network.*

We appreciate your feedback on the reliability of this figure. It appears to us that we face a tradeoff between completeness of the methodology description and ease of reading in this scheme. However, we would like to prioritize completeness and offer the reader a full comprehension of our methodology based on this figure. Furthermore, the explicit representation of the DEVINE model following codes from Le Toumelin et al. 2023 helps readers to the original DEVINE publication to this work.

- P13 L308: steps --> step

Corrected.

- P14 first paragraph: If TPI and elevation are strongly correlated, it makes no sense to compare their respective influences. Please compute the correlation, and, if it is high, I suggest to state that it is not possible to identify any distinction between the two in terms of influence on the result. You do note this to some extent, but this should be emphasized and could reduce the discussion of this paragraph significantly and rendering the corresponding figure moot.

The reviewer makes a good point. Our original formulation stating a high correlation between TPI and elevation could not fully support our analysis.

We computed the correlation between TPI and elevation of the observation station (Pearson correlation coefficient) and it equals to 0.39 (0.58 when we compare the absolute value of TPI to elevation). In other words, TPI and elevation are partly correlated.

Our intuition behind this result is that elevated regions (represented by the variable elevation) host more peaks and ridges (for which TPI is a good proxy). Similarly, flat terrain is often found on low elevation valleys. However, we can find peaks and ridges at lower elevation, as well as plateaus on elevated terrain which explains the partial correlation between both variables.

In our opinion, the part of our analysis dedicated to the joint effect of TPI is valid (even though not optimally formulated): the negative bias of simulated wind speed increases for the highest stations in our dataset when TPI increases. Reversely, for the highest TPI, elevation will increase the negative bias. The intuition is that elevation and summit act on wind speed through two distinct processes: high elevation wind fields encounter the surface more easily at elevated stations (hence

increasing observed speed and potentially increasing simulated negative bias). Venturi effect permits the acceleration of observed wind speed on summits and ridges. We believe these two processes are independent and can add on top of each other.

We clarified and simplified the text by specifying the correlation value and removing sentences that exaggerated the relationship between TPI and elevation. The new text is (found L330 in the first version of the manuscript):

*Notably, we observe that elevated stations are partially correlated with the TPI_500m (Pearson correlation coefficient = 0.39). High positive values of TPI_500m indicate that the observation station dominates its neighborhood, and is to some extent "exposed". TPI_500m close to zero characterize stations on average at the same elevation as their neighborhood in a radius of 500m, a definition that includes flat terrain.*

- Figure 5: I am unfamiliar with wind roses (though I expect most readers won't be). Does the size of the spoke indicate the frequency with which the direction in the correspond bin occurs for all panels? So only the color coding is different than the rest for panel d (observations)?

In this figure the spoke indicates the proportion of data in the considered direction. For instance, Figure 5 (a) indicates the most frequent wind direction simulated by AROME_forecast in the South-West (S-W) direction. According to the number on the radial axis, the frequency of wind direction lying in this range is slightly superior to 10.4 %.

The colorbar indicates the modeling error. Following the same panel (Figure 5, (a)), most directions forecasted in the South-West direction present errors lower than 30° (dark blue).

To clarify this point, we changed:

*"Numbers on the radial axis indicate the proportion in % of data in each bin compared to the whole dataset."*

into:

*"The spoke on the radial axis indicates the proportion in % of data that is predicted in the considered direction."*

- P15 L350: Please define RMSE and MAE (not just in figure/table captions)

The definition of MAE is now written in this sentence. RMSE definition is stated at the first occurrence of the word L271.

- P16 L366: reinforce previous --> reinforce a previous

Corrected.

- P16 L369: elevations --> elevation

Corrected.

- P16 L375: (Fig 5) However --> (Fig 5). However

Corrected.

- P20 L426: observation --> observations

Corrected.

- I really like paragraph 4.5. It is indeed emphasizing the importance of separating the corrective step from the downscaling step!

We thank the reviewer for this positive feedback.

- P22 L457: modification --> modifications

Corrected.

- P22 L458: Related to comments on P3 L72 and P10 L229, I don't understand how the network is aware of anything related to DEVINE.

We refer to our previous comment concerning the optimization procedure. As a consequence, we rephrased L458 to:

*Since the optimization of Neural Network has been obtained after backpropagating error gradients through DEVINE and both ANNs, we can expect that the deep learning model is to some extent aware of the expected effect of DEVINE and prevents from overcorrecting AROME_forecast.*

- P22 L467: application --> applications

Corrected.

- P23 L481: I don't understand this sentence. What challenges? Do you mean the evaluation is more critical? If yes, please rephrase. Otherwise, please explain.

We thank the reviewer for noticing this unclear sentence. We have rephrased the sentence as follows:

*This corresponds to a very strict evaluation procedure, which makes it generally harder to obtain good evaluation metrics versus simpler evaluation procedures that only perform tests at the sites included in the training set (Bolibar et al., 2020; Dujardin and Lehning, 2022).*

- Figure 10: Do I understand correctly that ALE are approximated local gradients around the value of interest, averaged over time and space? And the shaded region represents to corresponding standard deviation? Are the solid lines (the means) also accumulated?

Accumulated local effects (ALE) are tools to describe how a single input variable affects the output on average. The difference between this tool and comparable alternatives (e.g. Partial Dependence Plots) is that ALE give robust estimates of the average effect on the output even though input variables are strongly correlated one to each other (as it is in our case).

ALE work by first gridding the distribution of an input variable into a discrete interval. In our case we used quantiles to build the interval. Then, we select all data instances with the input variable of interest for a given bin in the grid. We alternatively modify the instances by setting the variable of interest to the value of the uppermost value of the bin. Then, we make a model prediction. We do the same procedure but replace the input variable of interest with the lowermost value of the bin. We then compute the difference between the results. These steps can be interpreted as a partial derivative estimation with respect to the input variable of interest (we referred to a gradient estimation even though it's not formally the same). This is the "error" in accumulated local errors.

We then average the differences obtained in the following step for each bins. Averaging model outputs for given data instances induces averaging outputs obtained at different time steps and locations. This corresponds to the term "local" in ALE.

Then, we accumulate the error and obtain the solid lines. This step can be interpreted as an integration of the partial derivative of the results with respect to a defined input variable. This corresponds to "accumulated" in ALE. In the end, this procedure permits to isolate the effect of the input variable of interest on the outputs (through the "derivative") and then accumulate ("integrate") the results in order to retrieve the averaged effect of the input variable. This is now corrected L306:

*This step can be interpreted as a computation of a partial derivative around a specific value of variable_i*

Since this method is based on computation of averaged values, it is also important to note when averages hide large spread around mean values (i.e. very distinct individual effects of the input variable of interest among the different instances). For that, we computed standard deviations. These deviations are also accumulated: since we accumulated mean values, deviations to the means are also accumulated as a way to illustrate the potential spread increase.

Finally, the solid line in the plot corresponds to the ALE, and the shaded regions the accumulated spread following the computation of averages in the ALE.

We summarize this discussion by adding the following information to Sect. 3.5.2 L 306:

*The differences are then averaged to obtain the local effect of variable_i for the considered bin. A standard deviation around the mean value is also computed, as a way to track the dispersion of individual effects. Local effects are then accumulated and centered across each bin to finally obtain ALE. This step corresponds to an*

*integration of the (averaged) local gradients and enables to represent the dependence of model outputs to variable_i across its range. In this study, we also accumulated the standard deviations as a way to keep track of the dispersion characterizing the individual effects (shaded regions in Fig. 10).*

We now specify that ALE correspond to the solid lines in the caption of Figure 10:

*Accumulated Local Effect (ALE) associated with each input variable of ANN_speed (solid lines).*

- P25 L524: I don't see any indication of skip connections in Fig2. Perhaps it is because the lines are too thin, or the figure too small (also see previous comment about Fig 2).

Following our answer to your previous comment, the skip connections are now mentioned in the caption.

They are highlighted by the connection that bypasses the fully connected network on the left part of Fig. 2. They are also indicated by the "+" sign. This is now indicated in the caption.

- P25 L528: What exactly is the difference between real elevation and model elevation?

Real elevation is the real elevation above sea level of the observation station. Model elevation is the elevation of the NWP grid cell that gives meteorological information to our model.

Please find below a short explanation on why model elevations differ from real elevations:

NWP models numerically integrate differential equations representing the state of the flow using numerical methods based on a grid discretisation. Grids are characterized by their horizontal grid spacing, which is often on the order of one to several kilometers for current NWP models (1.3 km for AROME). As a consequence, NWP represents elevation as a variable on the discretized grid. Given the coarse scale resolution of NWPs, it is very common that "model elevation", i.e. elevation as considered in the NWP grid, differs from "real elevation".

- P25 L535: Again, I am confused by this. How had neural network seen downscaled simulations?

We refer to our previous comments concerning the effect of DEVINE on the parameters of the neural networks during training.

- P27 L589: I would delete the i) and ii). At least for me it was only confusing.

We deleted the ii) and (ii).

- P28 L595: "best results" should be replaced with something like "most improvements".

  Corrected.

References:

Quéno, L., Vionnet V. , Dombrowski-Etchevers I. , Lafaysse M. , Dumont M. , and Karbou F. , 2016: Snowpack modelling in the Pyrenees driven by kilometric-resolution meteorological forecasts. *Cryosphere*, **10**, 1571–1589, doi:10.5194/tc-10-1571-2016.

Vionnet, V., Dombrowski-Etchevers, I., Lafaysse, M., Quéno, L., Seity, Y., & Bazile, E. (2016). Numerical weather forecasts at kilometer scale in the French Alps: Evaluation and application for snowpack modeling. *Journal of Hydrometeorology*, *17*(10), 2591-2614.

Gouttevin, I., Vionnet, V., Seity, Y., Boone, A., Lafaysse, M., Deliot, Y., & Merzisen, H. (2023). To the origin of a wintertime screen-level temperature bias at high altitude in a kilometric NWP model. *Journal of Hydrometeorology*, *24*(1), 53-71.

Le Toumelin, L., Gouttevin, I., Helbig, N., Galiez, C., Roux, M., & Karbou, F. (2023). Emulating the Adaptation of Wind Fields to Complex Terrain with Deep Learning. *Artificial Intelligence for the Earth Systems*, *2*(1), e220034.